# SI-traceable validation of a laser spectrometer for balloon-borne measurements of water vapor in the upper atmosphere

**Simone Brunamonti[1], Manuel Graf[1], Tobias Bühlmann[2], Céline Pascale[2], Ivan Ilak[1], Lukas Emmenegger[1], and Béla Tuzson[1]**

[1]Empa – Swiss Federal Laboratories for Materials Science and Technology, Laboratory for Air Pollution/Environmental Technology, Dübendorf, Switzerland
[2]METAS – Swiss Federal Institute of Metrology, Laboratory Gas Analysis, Berne-Wabern, Switzerland

**Correspondence:** Simone Brunamonti (simone.brunamonti@empa.ch) and Béla Tuzson (bela.tuzson@empa.ch)

**Abstract.** Despite its crucial role in the Earth's radiative balance, upper-air water vapor ($H_2O$) is still lacking accurate, in situ, and continuous monitoring. Especially in the upper troposphere–lower stratosphere (UTLS), these measurements are notoriously difficult, and significant discrepancies have been reported in the past between different measuring techniques. Here, we present a laboratory assessment of a recently developed mid-IR quantum-cascade laser absorption spectrometer, called ALBATROSS, for balloon-borne measurements of $H_2O$ in the UTLS. The validation was performed using SI-traceable reference gas mixtures generated based on the permeation method and dynamic dilution. The accuracy and precision of ALBATROSS were evaluated at a wide range of pressures (30–250 mbar) and $H_2O$ amount fractions (2.5–35 ppm), representative of the atmospheric variability in $H_2O$ in the UTLS. The best agreement was achieved by implementing a quadratic speed-dependent Voigt profile (qSDVP) line shape model in the spectroscopic retrieval algorithm. The molecular parameters required by this parameterization were determined empirically using a multi-spectrum fitting approach over different pressure conditions. In the laboratory environment, ALBATROSS achieves an accuracy better than $\pm 1.5\%$ with respect to the SI-traceable reference at all investigated pressures and $H_2O$ amount fractions. The measurement precision was found to be better than 30 ppb (i.e., 0.1 % at 35 ppm $H_2O$) at 1 s resolution for all conditions. This performance, unprecedented for a balloon-borne hygrometer, demonstrates the exceptional potential of mid-IR laser absorption spectroscopy as a new reference method for in situ measurements of $H_2O$ in the UTLS.

## 1 Introduction

Water vapor ($H_2O$) is the strongest greenhouse gas in the Earth's atmosphere and is a major driver of the atmospheric dynamics, microphysics, and interaction with radiation (IPCC, 2021). Its abundance in the atmosphere decreases strongly with altitude, from typically 1 %–3 % close to the surface to around $5\,\mu mol\,mol^{-1}$ (or parts per million, ppm) in the stratosphere. Despite its scarcity, upper-air $H_2O$ is still of great importance to the radiative balance of the atmosphere. Particularly in the upper troposphere–lower stratosphere (UTLS), i.e., at altitudes of about 8–25 km, even small changes in $H_2O$ were shown to have a significant impact on the rate of global warming (Solomon et al., 2010; Riese et al., 2012). Hence, accurate measurements of $H_2O$ in the UTLS are crucial for reliable climate predictions. However, due to its low amount fractions and the low temperatures ($T < -60\,°C$) of the UTLS, accurate measurements of $H_2O$ in this region are notoriously difficult.

Since the pioneering work by Brewer and Dobson (1951), a large amount of scientific research has been done on the water vapor distribution and variability in the upper atmosphere, based on a wide range of platforms and analytical techniques (e.g., Scott et al., 1999; Rosenlof et al., 2001; Gurlit et al., 2005; Sargent et al., 2013; Buchholz et al., 2014; Meyer et al., 2015). Several studies comparing in situ measurements

of UTLS $H_2O$ from both aircraft and balloon platforms found significant discrepancies between different measuring techniques (e.g., Oltmans et al., 2000; Vömel et al., 2007; Rollins et al., 2014; Brunamonti et al., 2019; Singer et al., 2022), with implications for the understanding of ice microphysical processes (e.g., Peter et al., 2006; Krämer et al., 2009). Large relative discrepancies (exceeding $\pm 100\%$) between different hygrometers were also reported from laboratory experiments simulating the UTLS conditions, such as AquaVIT-1 (Fahey et al., 2014, and references therein), where only a small subset of instruments were able to achieve mean deviations below $\pm 10\%$ from a reference value in all conditions.

Currently, cryogenic frost-point hygrometry (CFH) is considered to be a state-of-the-art method for balloon-borne measurements of UTLS $H_2O$ (Fahey et al., 2014) and is routinely used in global long-term monitoring networks, such as the GCOS Reference Upper-Air Network (GRUAN) (e.g., Hurst et al., 2011). CFH instruments are based on the chilled-mirror principle and have an estimated uncertainty of 4 %–6 % in the UTLS (Hall et al., 2016; Vömel et al., 2016). However, these devices are currently undergoing a fundamental reconception because the cooling agent fluoroform (HFC-23) used for their operation must be phased out due to its high global warming potential (UNEP, 2016). Thus, there is an urgent need for alternative, reliable technologies for the long-term monitoring of UTLS $H_2O$. Alternative measurement techniques demonstrated for lightweight balloon platforms include Lyman-$\alpha$ fluorescence (Zöger et al., 1999; Sitnikov et al., 2007; Khaykin et al., 2013) and laser absorption spectroscopy (Durry et al., 2008; Graf et al., 2021).

The aim of this work is to validate the accuracy and precision of a newly developed open-path, mid-IR quantum-cascade laser absorption spectrometer (ALBATROSS; Graf et al., 2021). This direct absorption-based technique is promising in that it can be a calibration-free method (Buchholz and Ebert, 2018), which makes it exceptionally attractive for demanding field applications. In practice, however, the accuracy can suffer from uncertainties in physical (pressure, temperature) as well as molecule-specific spectroscopic parameters, which will ultimately limit the quality of the retrieved data. While the error contribution of the environmental factors can be estimated based on the measurement uncertainties of these quantities, their large variations encountered in the stratosphere will also impact the intrinsic molecular properties of the absorption line, such as the line strength, its temperature dependence, and the pressure-broadening parameters. Detailed knowledge of the latter is in particular a prerequisite for an accurate spectral retrieval. Our focus is on the broadening effects, while for the line strength and its temperature dependency we take the values from the HITRAN2020 database (Gordon et al., 2022).

Here, we address this aspect by carrying out a dedicated laboratory investigation conducted at the Swiss Federal Institute of Metrology (METAS). Using a dynamic–gravimetric permeation method (e.g., Haerri et al., 2017; Guillevic et al., 2018), we generated SI-traceable $H_2O$ amount fractions as low as 2.5 ppm in synthetic air with an expanded measurement uncertainty smaller than $\pm 1.5\%$. This allowed us to improve our spectroscopic retrieval algorithm by implementing a more advanced line shape model than the standard Voigt profile, i.e., the quadratic speed-dependent Voigt profile (qSDVP), and to then assess the accuracy and precision of ALBATROSS at a wide range of UTLS-relevant conditions. The molecular parameters required by the qSDVP parameterization were determined experimentally using a multi-spectrum fitting approach (e.g., Cygan and Lisak, 2017).

## 2 Experimental setup

### 2.1 ALBATROSS spectrometer

The ALBATROSS spectrometer, described in detail in Graf et al. (2021), leverages recent advances in optics and laser driving concepts. It incorporates a monolithic, segmented circular multipass cell (SC-MPC), consisting of a rotationally symmetric arrangement of individual mirror segments carved into its inner surface (Graf et al., 2018). This geometry was found to be highly tolerant to thermally induced distortion, robust to mechanical stress, and thus well suited for open-path applications (Tuzson et al., 2020). The SC-MPC contains 57 mirror segments ($6 \times 6\,mm^2$) with a diagonal distance of 108.8 mm, resulting in an effective optical path length (OPL) of 610.7 cm. The last segment of the SC-MPC is designed such that the laser beam is directly focused onto the IR detector without any additional beam-shaping optics. A distributed feedback quantum-cascade laser (DFB-QCL), packaged in a high-heat-load housing with embedded thermoelectric cooling and collimation optics (Alpes Lasers SA, Switzerland), is used as a light source. The laser is tuned across a $\sim 1\,cm^{-1}$ spectral window centered around an isolated $^1H_2^{16}O$ absorption line at 1662.809 $cm^{-1}$ ($\lambda \approx$ 6.01 µm). This absorption line corresponds to the transition $221 \leftarrow 212$ with a lower-state energy of 79.5 $cm^{-1}$, which makes it largely insensitive to temperature. For example, a temperature change of 1 K would give $< 0.13\%$ change in the observed absorption amplitude. Since we take this effect into account in our calculations, and because the ambient temperature is kept constant, the impact of this term can safely be considered negligible. Furthermore, the estimated uncertainty of the line strength for the selected transition is known, according to the HITRAN database, with an accuracy better than 2 %.

A thermoelectrically cooled MCT detector (PVM-2TE-8 $1 \times 1$, Vigo Systems, Poland) is used for the detection of the transmitted light. Rapid spectral sweeping of the QCL is achieved by periodic modulation of the laser driving current, following the intermittent continuous-wave (ICW) modulation approach (Fischer et al., 2014). In this approach, the laser driving current is applied in pulses (typically 100 µs

long), followed by a short period of complete shutdown of the QCL. The ICW driving is obtained using custom-developed analogue electronics (Liu et al., 2018). The spectra are acquired at a frequency of 3 kHz and co-averaged to a resolution of 1 Hz. A full description of the laser driving and data acquisition systems can be found in Graf et al. (2021). The spectrometer weighs 3.41 kg, including batteries and a Styrofoam thermal insulation enclosure.

For its laboratory operation, ALBATROSS was set up into a closed-path configuration, in which the SC-MPC is closed on both sides by stainless-steel lids. The lids, as well as all tubes in contact with the gas, were treated by a highly inert coating (SilcoNert® 2000, SilcoTek, USA) that minimizes the adsorption of molecules on its surface (e.g., Vaittinen et al., 2018; Macé et al., 2022), thereby minimizing memory effects and shortening the response time. The required leak tightness was achieved by using nitrile O rings (between the cell body and the lids) and high-vacuum metal fittings (VCR, Swagelok, USA) for the gas handling system. The SC-MPC volume in a closed-path configuration is about 140 cm$^3$. Furthermore, the free-space path between the key optical elements, i.e., laser MPC detector (kept by design as short as physically possible, in our case, 2.7 cm), was enclosed by a custom-made flexible PEEK-polymer tubing that was purged with dry $N_2$ and was maintained slightly above atmospheric pressure to avoid any "parasitic" $H_2O$ absorption from these external path sections.

## 2.2 Reference gas generation

The SI-traceable reference gas mixtures used for the validation of ALBATROSS were generated based on the permeation method combined with dynamic dilution (following ISO 6145-7 and ISO 6145-10). This is an established, standardized technique in metrology, particularly for reactive or polar compounds (e.g., Scaringelli et al., 1970; Brewer et al., 2011; Haerri et al., 2017; Guillevic et al., 2018). The setup used here is the same as described in Guillevic et al. (2018). All parts in contact with the gas mixture are passivated with SilcoNert® 2000. The mass loss of the permeator is determined by a magnetic suspension balance (MSB; Rubotherm, Germany) under controlled temperature and pressure. This allows for continuous and unperturbed mass measurements, as the permeator is physically decoupled from the balance. To correct for balance drift and buoyancy, two calibration masses traceable to the SI realization of the kilogram of Switzerland (Fuchs et al., 2012) are automatically placed on the balance plate at regular intervals during the calibration routine. The permeator used here (Fine Metrology, Italy) has a nominal permeation rate of approximately 10 µg min$^{-1}$ at 45 °C for $H_2O$ and was filled with ultrapure water (resistivity 18.2 MΩ cm at 25 °C, corresponding to a purity > 99.999 %). The expanded relative measurement uncertainty of the generated $H_2O$ amount fractions, determined upon calibration of the permeator, including its drift

and the dynamic dilution unit, was found to be better than ±1.5 % in all conditions.

As the isotopic composition of our water standard is not known, we estimate an additional uncertainty of about ±0.02 % in the total $H_2O$ amount fraction by assuming that the liquid water standard has the signature of typical tap water, i.e., $-60$‰ $\delta^2H$ and $-10$‰ $\delta^{18}O$. Considering the low abundance of the heavy isotopologues and a natural distribution, the total $H_2O$ amount fraction contains about 99.73 % of the light water isotopologue ($^1H_2^{16}O$). Since ALBATROSS only measures this water isotopologue species and not the total $H_2O$ amount fraction, the reference values generated by the permeation method were scaled by this factor for the accuracy assessment.

## 2.3 Gas handling system

Figure 1 shows a schematic of the dilution and sampling system used for the measurements. The carrier gas is synthetic air that is passed through a gas purifier (MicroTorr MC400, Saes, Italy), which reduces the $H_2O$ content to about 70 ppb (previously measured by cavity ring-down spectroscopy). The volumetric flow rates in the different branches of the dilution and sampling system are controlled by a series of mass flow controllers (MFCs). The humidity content of the reference gas flow is adjusted by varying the flow rate of the dry synthetic air through MFC 2 between 0.05 slpm (corresponding to $\sim$ 35 ppm $H_2O$) and 4.5 slpm ($\sim$ 2.5 ppm $H_2O$) to dilute the $H_2O$-enriched airstream that passes at a constant flow rate of 0.3 slpm (set by MFC 1) through the permeation chamber of the MSB. The total flow rate through the SC-MPC is kept constant at 0.3 slpm by MFC 3, while a solenoid valve allows the automatic switching between the reference gas mixture and the zero air (i.e., dry synthetic air) supplies. A vacuum pump and an upstream pressure controller (PC-15PSIA, Alicat, USA) are used to set the pressure in the multipass cell at various levels between 30 and 250 mbar. The sample pressure in the SC-MPC was monitored by a heated capacitance manometer (AA02, MKS Instruments, USA), with an absolute accuracy of 0.12 %. The standard deviation of the measured pressure was within 0.1 % during all experiments. All measurements were performed in an air-conditioned laboratory, and the spectrometer was operated inside a custom-made PMMA plastic chamber (thickness 15 mm) to further suppress any sudden temperature or humidity fluctuations. The temperature was monitored in the vicinity of the multipass cell (inside the PMMA chamber) by an HMP110 sensor (Vaisala, Finland), with an absolute accuracy of 0.2 °C. The average temperature measured during the campaign period was $23.5 \pm 0.03$ °C.

In addition to the SI-traceable reference gas mixtures, another series of measurements was performed using a secondary reference gas mixture. This allowed us to extend the range of the validation beyond the maximum value of 35 ppm $H_2O$. The custom-made secondary reference gas was pro-

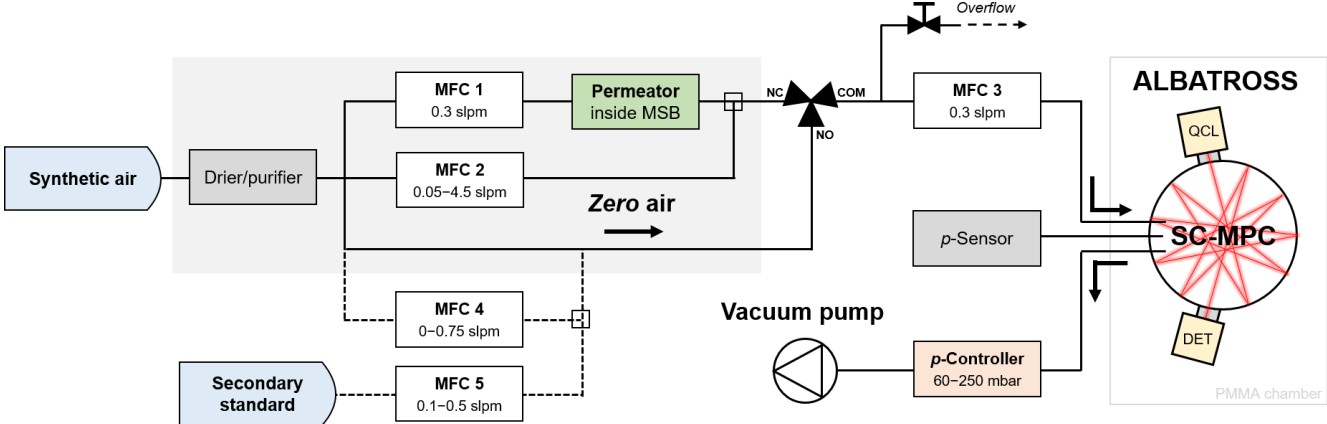

**Figure 1.** Scheme of the sampling system used for the validation measurements. The grey shaded area indicates the magnetic suspension balance (MSB) and the dilution system used for the generation of the SI-traceable reference gases.

duced by spiking a gas cylinder of synthetic air with a known amount of pure water. For this purpose, a regular steel cylinder (previously containing synthetic air) was evacuated and filled with synthetic air up to 1 bar, and then a syringe with distilled water was used to introduce a given amount of $H_2O$, followed by pressurizing the gas cylinder with synthetic air (with $< 5$ ppm $H_2O$ content; Messer, Switzerland) up to 100 bar, resulting in a humidity content of about 180 ppm $H_2O$. After a few days of equilibration, the gas was expanded into a 34 L SilcoNert® 2000-coated stainless-steel cylinder (Essex Industries Inc, USA) to further minimize any potential surface effects during the measurements.

It should be noted that this custom-made secondary reference gas does not fulfill SI traceability, and it is subject to well-known long-term stability issues. Its sole purpose is to assess whether ALBATROSS is capable of measuring significantly higher water vapor amounts than can be generated by the permeation method.

## 2.4 Measurement overview

The $H_2O$ amount fraction levels (or "set points") and gas pressure ($p$) levels were selected to be representative of the atmospheric profile of $H_2O$ in the UTLS. Figure 2a shows the distribution of all $H_2O$ set points and $p$ levels investigated in this work, overlaid with two atmospheric profiles of $H_2O$ amount fractions measured by CFH during recent field campaigns. The profiles correspond to moist, tropical summer conditions (black line) and dry, mid-latitude winter conditions (grey line).

We selected five $H_2O$ set points covering the full range offered by the dynamic–gravimetric reference gas generation source (2.5–35 ppm) and eight $p$ levels between 30 and 250 mbar for a total of 40 possible combinations (red stars in Fig. 2a). This allows us to fully cover the expected variability range in UTLS $H_2O$. The generated $H_2O$ amount fractions and selected $p$ levels, along with their relative uncer-

**Table 1.** Summary of the $H_2O$ amount fraction levels generated by the dynamic–gravimetric method and their expanded measurement uncertainty **(a)** and pressure levels as well as their measurement uncertainty **(b)** used for the absolute validation. The relative expanded measurement uncertainty in the $H_2O$ amount fraction levels varies between 1.4 % and 1.47 % for all conditions.

| **(a)** $H_2O$ amount fraction, reference levels | |
|---|---|
| Target value (ppm) | Actual value (ppm) |
| 2.5 | $2.514 \pm 0.037$ |
| 5 | $4.93 \pm 0.07$ |
| 10 | $9.84 \pm 0.14$ |
| 20 | $20.05 \pm 0.28$ |
| 35 | $34.58 \pm 0.51$ |

| **(b)** Pressure levels | |
|---|---|
| Target value (mbar) | Actual value (mbar) |
| 30 | $31.52 \pm 0.04$ |
| 45 | $45.14 \pm 0.05$ |
| 60 | $61.13 \pm 0.07$ |
| 80 | $80.67 \pm 0.10$ |
| 100 | $101.55 \pm 0.12$ |
| 150 | $151.75 \pm 0.18$ |
| 200 | $201.96 \pm 0.24$ |
| 250 | $251.62 \pm 0.30$ |

tainties, are listed in Table 1. The conditions generated using the secondary reference gas mixtures for the extended-range validation (blue stars in Fig. 2a) included five $H_2O$ set points (between approximately 22 and 180 ppm) and three $p$ levels (60, 100, 200 mbar). These measurements allow us to assess the linearity of ALBATROSS under conditions relevant for the upper troposphere, i.e., roughly up to 6–10 km altitude.

Figure 2b shows a representative time series of the measurement performed at 150 mbar. Each $H_2O$ set point generated by the dynamic–gravimetric method is measured for 2 h

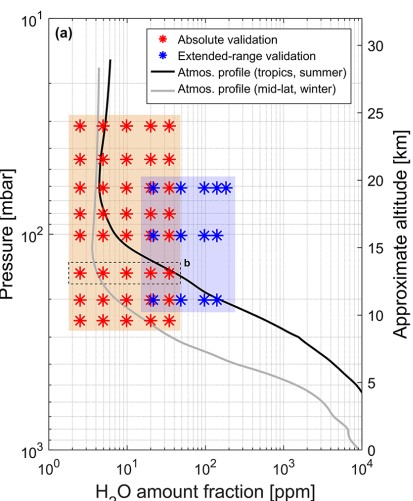
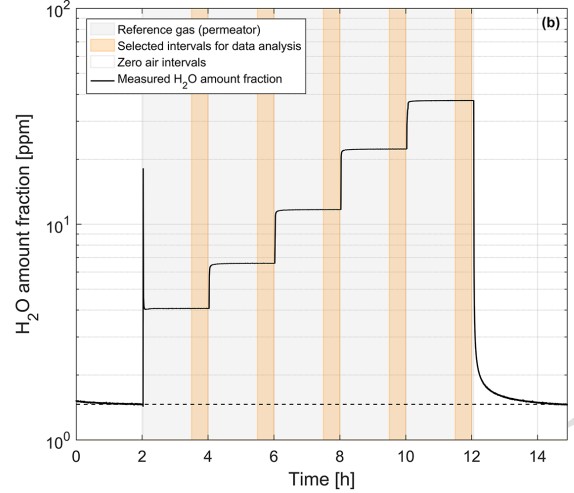

**Figure 2.** Overview of the laboratory validation measurements. **(a)** Experimental settings in terms of pressure ($p$) and $H_2O$ amount fraction levels investigated for the absolute validation (red stars, SI-traceable reference) and the extended-range validation (blue stars, secondary reference) measurements, overlaid with two atmospheric profiles of UTLS $H_2O$. The profiles correspond to tropical summer conditions (black: Brunamonti et al., 2018) and mid-latitude winter conditions (grey: Graf et al., 2021) and were smoothed by a $\pm 1$ km moving average. **(b)** Time series of the $H_2O$ amount fraction measured during the individual experiment at 150 mbar gas pressure, retrieved using the qSDVP line shape model at 1 s resolution. Different color shadings indicate zero air measurements (white), reference gas measurements (grey), and the 30 min intervals selected for the data analysis of each $H_2O$ level (orange). The estimated $H_2O$ amount fraction content of the zero air (1.46 ppm) is indicated by a dashed black line.

to allow the equilibration of the $H_2O$ amount fraction in the SC-MPC (Fig. 2b). The last 30 min of each interval is then selected for the precision assessment. Before and after each experiment, the SC-MPC is purged with zero air for at least
5 3 h to obtain the "empty-cell" spectrum, i.e., the transmission signal of the SC-MPC in the absence of the reference gas. This is used to normalize the subsequent raw absorption spectra. Despite the substantial purging by dry zero air, the lowest measured $H_2O$ amount fraction in the SC-MPC was
10 about 1.5 ppm after 3 h of purging. This behavior is mainly due to the strong surface adsorption/desorption properties of $H_2O$, causing a memory effect in the system. Throughout the validation experiments, we observed a tendency towards slightly elevated zero levels whenever the gas pressure in the
15 sampling line was lowered. Furthermore, the response time of the instrument showed a clear correlation with the humidity content of the measured gas.

While these effects may, if not properly taken into account, e.g., by the empty-cell spectrum normalization, affect the
20 accuracy of the measurements, they are largely absent during flight conditions, where the instrument is operated in an open-path configuration. In this latter case, there is no sampling line, the gas flow is much larger, and the surfaces of the SC-MPC are drastically reduced as the lids are removed
25 and the gas–surface interaction is limited to the narrow inner circumference of the cell.

## 3 Data analysis

### 3.1 Spectroscopic retrieval

The $H_2O$ amount fractions are retrieved from the measured spectra using the Lambert–Beer law in combination with 30 the ideal gas law. Molecular line parameters are taken from the HITRAN2020 database (Gordon et al., 2022), and the wavenumber dependency of the absorption coefficient is approximated by a line shape model. Typically, the Voigt profile (VP), a convolution of Gaussian (Doppler-broadening) 35 and Lorentzian (pressure-broadening) profiles, is considered to be a good compromise to capture the pressure-broadening effects while offering a good stability and reliability for airborne hygrometers (e.g., Buchholz et al., 2014, 2017; Graf et al., 2021). However, there is growing experimental evidence 40 that various non-Voigt effects (e.g., Dicke narrowing, line narrowing due to the speed dependence of pressure-induced broadening, and line asymmetry caused by the speed dependence of pressure shift) can impact the observed absorption line profile (e.g., Hodges et al., 2008; Kochanov, 2012; Ngo 45 et al., 2012; Tennyson et al., 2014; Lisak et al., 2015). Thus, one major aim of this work was to identify the line shape model that provides the best accuracy and reliability throughout the entire range of UTLS conditions.

Recently, the Hartmann–Tran profile (HTP) has been recommended as a new standard in spectroscopic databases (Tennyson et al., 2014). However, this model requires a large number of line-specific parameters, which are difficult to de-

termine considering the correlations between them and the relatively moderate signal-to-noise ratio (SNR) of our measured spectra. Therefore, we investigated lower-order models that could still reproduce the measured spectra at high fidelity (i.e., to nearly the experimental noise level). Our analysis revealed the quadratic speed-dependent Voigt profile (qSDVP), in which the speed dependence of the relaxation rates is considered the sole source of line broadening, to be the most suitable.

The qSDVP model is characterized by the phenomenological rate parameters $\Gamma_2$ and $\Delta_2$ to describe the quadratic dependence on the active-molecule speed of the pressure-broadening width and shift, in addition to the collisional width and shift ($\Gamma_0$ and $\Delta_0$) averaged over all speeds (Tennyson et al., 2014). While the molecular parameters required by the VP line shape model can be readily obtained from the HITRAN2020 database, the corresponding qSDVP parameters for the $H_2O$ transition used here are not available in the literature. Hence, we had to determine these parameters experimentally. The parameter optimization procedure is discussed in detail in Sect. 3.3–3.4.

Once the molecular parameters are defined, the $H_2O$ amount fractions are retrieved by minimizing the squared differences between the measured spectra and the line shape model (i.e., the fitting residuals), using a Levenberg–Marquardt least-squares algorithm (Press et al., 2007). The overall measurement uncertainty in the amount fractions retrieved by this method, associated with the uncertainty in the measured environmental parameters that are used in the calculation, is estimated to be less than 0.2 %. This results from combining the absolute uncertainties in the measured $p$ (0.12 %) and $T$ (0.06 %) and their variability throughout the measurement periods (standard deviations 0.05 % and 0.01 %, respectively). The uncertainty in the measured OPL (< 1 mm) contributes by less than 0.01 %.

## 3.2 Pre-processing

Prior to the spectroscopic retrieval, the raw spectra are pre-processed. This includes a normalization step, which can be done either by dividing each raw spectrum by the corresponding empty-cell spectrum (i.e., the transmission through the multipass cell filled with zero air) or by reconstructing the laser intensity baseline using a polynomial function (Graf et al., 2021). Then, the time domain is converted to the frequency domain using the transmission spectrum of a 5.08 cm. long Ge etalon, and the data are interpolated to an equally spaced wavenumber grid. For this, a free spectral range (FSR) value of 0.02429 cm$^{-1}$ was determined by optimizing the retrieved peak position of two neighboring absorption lines of $N_2O$ (measured at 100 mbar) to their line positions given in HITRAN2020 (see Fig. S1 in the Supplement). The spectral range covered by the QCL is 0.88 cm$^{-1}$. The number of data points per spectrum is reduced by a factor of 4, i.e., from $2.1 \times 10^4$ to $5 \times 10^3$, by using the moving

average approach, resulting in a uniform spectral-point resolution of $1.67 \times 10^{-4}$ cm$^{-1}$.

Figure 3 shows an overview of the measured spectra, normalized using the empty-cell transmission approach at $H_2O$ amount fractions of 2.5, 10, 35, and 140 ppm. These spectra were obtained by co-averaging 50 s of data, corresponding to a total number of $1.5 \times 10^5$ individual spectra. This choice is justified by the Allan–Werle deviation analysis, discussed in Sect. 4.1. The SNR, defined as the ratio of the peak absorption signal to the standard deviation of the measured spectrum, calculated in a frequency interval that excludes the absorption line center, is about 50 at 2.5 ppm $H_2O$ (Fig. 3a) and 2000 at 140 ppm $H_2O$ (Fig. 3d).

As the empty-cell spectrum can only be used in a closed-path configuration, we also tested the polynomial-baseline normalization approach for the case of open-path configuration of ALBATROSS, which is suitable for the evaluation of flight data. Therefore, we apply both normalization methods, and we show in the Supplement that they provide the same results in terms of accuracy of the retrieved $H_2O$ amount fractions (Fig. S2).

## 3.3 Voigt profile (VP)

As mentioned above, we first evaluated the performance of the VP model using the molecular parameters provided by the HITRAN2020 database, as well as by optimizing its parameters to obtain the best quality of the fit (QF) and agreement with the SI-traceable reference. However, neither approach was found to provide a sufficient level of accuracy or a good QF. Here, the QF is defined, following Cygan et al. (2012), as the ratio of the peak absorption signal to the standard deviation of the fit residuals, calculated over the entire spectrum. This definition accounts not only for the random noise in the experimental spectrum, as in the case of SNR, but also for systematic distortions caused by limitations in the line shape model (Cygan et al., 2012).

The optimization of the line profile parameters was performed using a multi-spectrum fitting (MSF) approach, in which spectra recorded at different pressures are fitted simultaneously by the least-squares algorithm (e.g., Cygan and Lisak, 2017). This method has the advantage of eliminating the partial correlations between the different line profile parameters that can occur in least-squares fits with single-pressure spectra. The procedure was implemented in Python using the HITRAN Application Programming Interface (HAPI) routines (Kochanov et al., 2016) as the core engine for the spectral fitting.

The performance of the VP was evaluated based on the difference in the $H_2O$ amount fraction between the spectroscopic retrievals and the SI-traceable reference values ($\Delta H_2O$), as well as the QF index. Figure 4 illustrates the VP optimization process, in terms of $\Delta H_2O$ (panel a) and the QF index (panel b) calculated from fitting the spectra recorded at 35 ppm $H_2O$ and eight different pressures (as

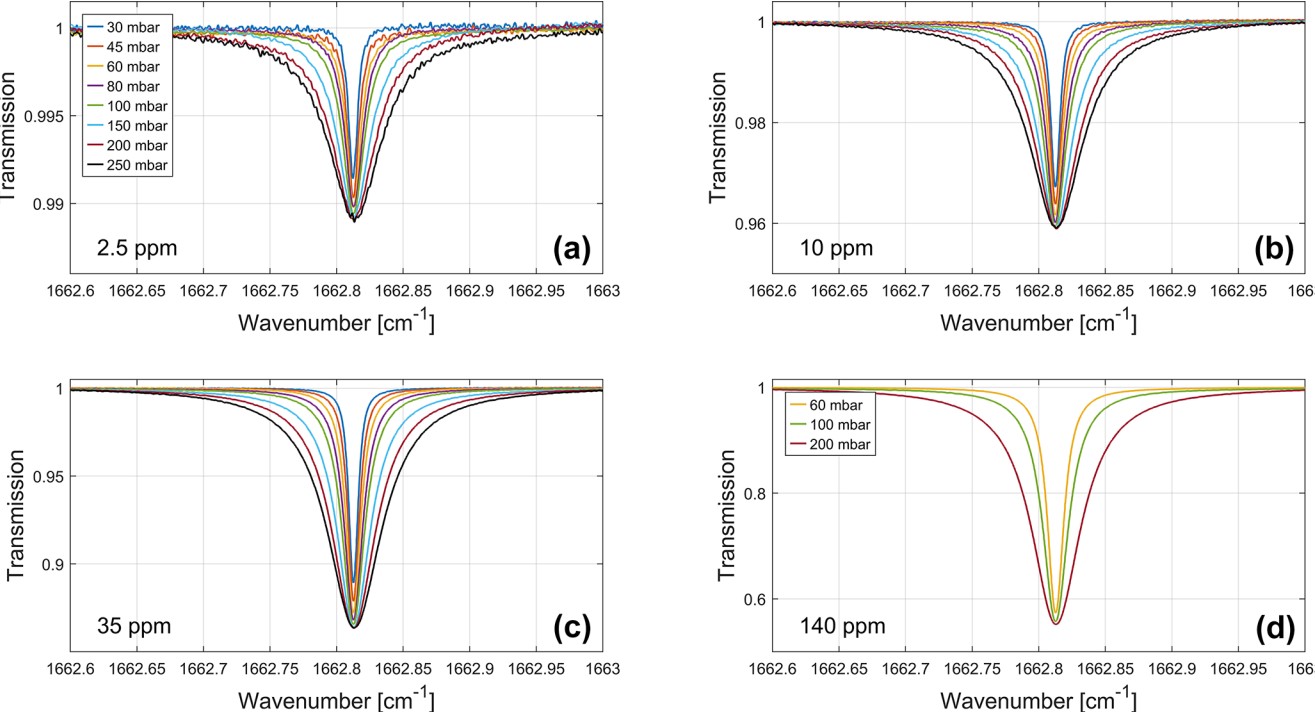

**Figure 3.** Normalized transmission spectra measured during the laboratory validation experiments at 2.5 ppm **(a)**, 10 ppm **(b)**, 35 ppm **(c)**, and 140 ppm $H_2O$ **(d)**, color-coded with gas pressure. The SI-traceable reference gas mixtures **(a–c)** were measured at eight pressure levels (30–250 mbar), while secondary reference gas mixtures **(d)** were measured at three pressure levels (60–200 mbar). All spectra shown here are integrated over 50 s.

shown in Fig. 3c), as a function of the pressure-broadening coefficient ($\Gamma_0$). This was obtained by using prescribed values of $\Gamma_0$ in the least-squares algorithm. Dashed vertical lines indicate the $\Gamma_0$ values from the HITRAN2020 database, namely $0.1002\,\mathrm{cm^{-1}\,atm^{-1}}$ at standard temperature (296 K) and pressure (1 atm) (STP conditions), and the value obtained by the MSF approach applied to the spectra measured at 35 ppm $H_2O$ and pressures 30–250 mbar ($\Gamma_0 = 0.0925\,\mathrm{cm^{-1}\,atm^{-1}}$ at STP). All the line shape parameter values used for the retrieval are summarized in Table 2.

As a first step, the retrieval was performed using the $\Gamma_0$ and line strength ($S_{ij} = 8.63 \times 10^{-20}\,\mathrm{cm^{-1}\,(molecule\,cm^{-2})^{-1}}$ at 296 K) parameters specified in HITRAN2020. However, these settings result in an overestimation of the retrieved $H_2O$ amount fractions by up to 5 % compared to the reference value (Fig. 4a) and a substantial spread of the deviations obtained at different pressures, indicating an increasing bias correlated with pressure (also shown in Fig. 8). Therefore, we used the MSF algorithm to optimize the $\Gamma_0$ value over the eight spectra considered here. This leads to the maximum of the QF index (Fig. 4b), but in this case, the retrieved $H_2O$ amount fractions underestimate the reference value by up to 9 % (Fig. 4a). Finally, we considered the case of varying $S_{ij}$, as recent studies have reported slightly different values of this parameter compared to HITRAN2020 for the $H_2O$ transition considered here (Ptashnik et al., 2016; Conway et

al., 2020; Birk et al., 2017; Delahaye et al., 2021). In this case, we observe that by increasing the line strength parameter by 5 % compared to HITRAN2020 (consistent with Ptashnik et al., 2016), it is possible to obtain on average good agreement with the reference values.

The major drawback of this approach is that the calibration-free character of the retrieval is lost because the optimization of the line strength value relies on the assumption of a target $H_2O$ concentration value. Furthermore, the good agreement found for one particular pressure does not apply to each individual pressure level, and a pressure-related bias still affects the results (see spread of grey lines in Fig. 4a at $\Gamma_0 = 0.0925\,\mathrm{cm^{-1}\,atm^{-1}}$ at STP). Hence, we conclude that the VP model is unable to reproduce the observed spectral characteristics with sufficiently high accuracy.

## 3.4 Quadratic speed-dependent Voigt profile (qSDVP)

After showing the limitations of the VP model, we proceeded to improve our spectroscopic retrieval algorithm with the qS-DVP parameterization. To determine the required parameters, the MSF algorithm was applied to the spectra collected at 140 ppm $H_2O$ and pressure levels 60–200 mbar (shown in Fig. 3d). This $H_2O$ level was chosen to achieve the highest possible SNR (i.e., about 2000), which is critical for an accurate estimation of the higher-order line shape parame-

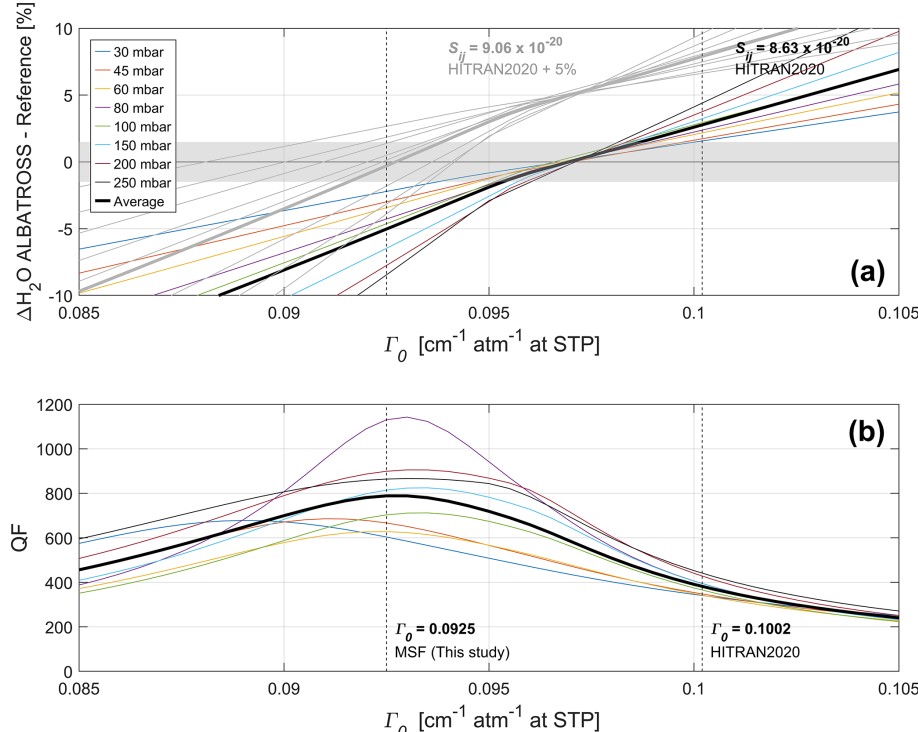

**Figure 4.** Voigt profile (VP) optimization. **(a)** Difference in the $H_2O$ amount fraction between the spectroscopic retrievals and the SI-traceable reference values ($\Delta H_2O$) at 35 ppm $H_2O$ and different pressures (color-coded) as a function of the pressure-broadening coefficient ($\Gamma_0$). The grey shaded area represents the relative uncertainty in the $H_2O$ reference levels generated by the permeator ($\pm 1.5\%$). The colored lines represent the results obtained using the line strength parameter ($S_{ij}$) from HITRAN2020, while grey lines represent the results obtained using a 5% higher $S_{ij}$ (given in units of $cm^{-1}$ (molecule $cm^{-2}$)$^{-1}$ at 296 K). **(b)** Quality of the fit (QF) index as a function of $\Gamma_0$, obtained from the VP model. Dashed vertical lines indicate the $\Gamma_0$ values obtained from the HITRAN2020 database ($0.1002\, cm^{-1}\, atm^{-1}$ at STP) and by the MSF approach ($0.0925\, cm^{-1}\, atm^{-1}$ at STP). In both panels, a thick black line shows the average value over all pressure levels.

**Table 2.** Summary of molecular parameters used for the spectroscopic retrieval. For the Voigt profile (VP) model, both the values taken from the HITRAN2020 database and those proposed in this study are reported. The quadratic speed-dependent Voigt profile (qSDVP) parameters were determined in this study, as discussed in Sect. 3. All values are expressed for standard temperature and pressure (STP) conditions (i.e., 296 K, 1 atm). * Note that a 5% higher $S_{ij}$ value was also tested for the VP optimization (Fig. 4), and slightly different estimates of $S_{ij}$ for this transition are also reported in the literature (see Sect. 3.3). The $\Gamma_2$ and $\Delta_2$ parameters are not applicable (n/a) to the VP parameterization.

| Line shape model parameters | | | |
|---|---|---|---|
| Parameter | Voigt profile (VP) | | qSDVP |
| | HITRAN2020 | This study | This study |
| $S_{ij}$ ($cm^{-1}$ (molec. $cm^{-2}$)$^{-1}$) | $8.63 \times 10^{-20}$ | $8.63 \times 10^{-20}$ * | $8.63 \times 10^{-20}$ |
| $\Delta_0$ ($cm^{-1}\, atm^{-1}$) | $3.87 \times 10^{-3}$ | $3.87 \times 10^{-3}$ | $3.87 \times 10^{-3}$ |
| $\Gamma_0$ ($cm^{-1}\, atm^{-1}$) | 0.1002 | 0.0925 | 0.0992 |
| $\Delta_2$ ($cm^{-1}\, atm^{-1}$) | n/a | n/a | 0 |
| $\Gamma_2$ ($cm^{-1}\, atm^{-1}$) | n/a | n/a | 0.0135 |

ters. To avoid overfitting, we fixed the $\Delta_0$ and $S_{ij}$ parameters to their values given in HITRAN2020 and set $\Delta_2 = 0$ (the reason for this is given below), while the most critical $\Gamma_0$ and $\Gamma_2$ parameters were determined by the MSF algorithm. As is shown below, this configuration is highly effi-

cient at fully reproducing the measured spectra at high fidelity and with high accuracy of the retrieved $H_2O$ amount fractions. The parameters resulting from the MSF calculation are $\Gamma_0 = 0.0992\, cm^{-1}\, atm^{-1}$ and $\Gamma_2 = 0.0135\, cm^{-1}\, atm^{-1}$ at STP.

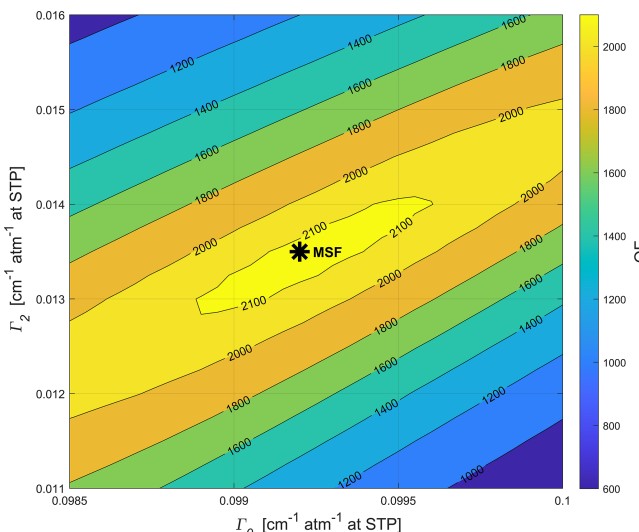

**Figure 5.** Determination of the qSDVP fitting parameters. Contour map of the quality of the fit (QF) index as a function of $\Gamma_0$ and $\Gamma_2$, obtained by fitting a qSDVP profile to the spectra recorded at 140 ppm $H_2O$. The $\Gamma_0$ and $\Gamma_2$ values obtained from the MSF approach applied to the same spectra ($\Gamma_0 = 0.0992\,cm^{-1}\,atm^{-1}$, $\Gamma_2 = 0.0135\,cm^{-1}\,atm^{-1}$ at STP) are indicated by a star. The QF index is calculated as the average QF over the three pressure levels measured at 140 ppm $H_2O$ (60, 100, and 200 mbar). The map was obtained by varying $\Gamma_0$ and $\Gamma_2$ in a regular $25 \times 25$ grid with a resolution of $6 \times 10^{-5}\,cm^{-1}\,atm^{-1}$ in $\Gamma_0$ and $2 \times 10^{-4}\,cm^{-1}\,atm^{-1}$ in $\Gamma_2$.

To illustrate that this approach provides a robust estimate of the pressure-broadening parameters, Fig. 5 shows a contour map of the QF index as a function of $\Gamma_0$ and $\Gamma_2$. This was obtained by consecutively varying $\Gamma_0$ and $\Gamma_2$ stepwise within a $25 \times 25$ grid, centered around their values obtained from the MSF method. The QF index shown here is calculated as the average QF over the three pressure levels measured at 140 ppm $H_2O$ (60, 100, and 200 mbar).

The results show that our MSF estimate of $\Gamma_0$ and $\Gamma_2$ lies in a well-defined maximum of the QF index (i.e., minimum of the standard deviation of the residuals), with values slightly exceeding 2000 (Fig. 5). This QF matches the SNR of the measured spectra, which is a strong indication that the line shape model reproduces the measured spectra to the experimental noise level. It is important to realize that, in this case, the line profile parameters are solely determined by the QF. The MSF algorithm is not aware of the target (or "true") value of the $H_2O$ concentration – it simply tries to minimize the sum of the squares of the residuals, i.e., the difference between the observed value and the fitted value provided by the model. Here, the model is based on first principles using the molecular parameters and the physical quantities ($p$, $T$, OPL). The generated SI-traceable $H_2O$ concentrations are only used for comparison purposes. There is no calibration involved.

Finally, Fig. 6 shows the fit residuals obtained using the VP (panels a–b) and qSDVP (panels c–d) line shape models, at 35 ppm $H_2O$ (panels a, c) and 140 ppm $H_2O$ (panels b, d). The VP retrieval is performed using the line parameters from the HITRAN2020 database, while for the qSDVP we use the $\Gamma_0$ and $\Gamma_2$ coefficients determined by the MSF method. Using VP yields systematic residuals significantly exceeding the measurement noise and exhibiting the characteristic W shape, with deviations as large as 1 % of the absorption signal near the line center (Fig. 6b). Conversely, using qSDVP, the fit residuals stay below $\sim 0.1$ % of the transmission signal for all pressures and $H_2O$ amount fractions (Fig. 6c–d). It can be tempting to include an additional line profile parameter, e.g., $\Delta_2$, to further reduce the remaining structures. However, our primary aim is to find an optimum compromise between establishing a reliable and accurate spectral retrieval and maintaining the high temporal (spatial) resolution of the spectrometer during balloon flights. The latter requires that we evaluate spectra from the flights at a 1 s acquisition time rather than averaging them over, e.g., 50 s. However, as the noise scales by $\sqrt{t}$ (assuming random fluctuations), its amplitude is about 7 times larger in the 1 s data compared to the situation shown in Fig. 6, and thus random-noise-induced statistical effects dominate the spectra. In our opinion, including another degree of freedom for the spectral fit under such circumstances is largely questionable. Another aspect is the consideration of various artifacts and their impact on the measured line profiles. As indicated in Fig. S1, the FSR determination uncertainty in our case is about $1.3 \times 10^{-5}\,cm^{-1}$. Furthermore, the frequency stability of our free-running QCL at longer timescales was found to be between 1.2 and $5.5 \times 10^{-5}\,cm^{-1}$ (mainly determined by the laser heat-sink temperature stability). While the former term affects the frequency-scale accuracy, the latter has a random bias on the line profile when averaging over multiple acquisitions. These influences can easily induce slight asymmetries or subtle line shape distortions that can then be erroneously assigned to the $\Delta_2$ parameter. Moreover, our trial of using $\Delta_2$ as a free fitting parameter in the MSF routine resulted in a weakly constrained value with large uncertainties, indicating difficulties of a proper assignment. Similarly, we also found that considering other parameters, e.g., collisional narrowing, does not improve the QF index. Therefore, we conclude, in full agreement with previous works (e.g., Lisak et al., 2015), that the observed line shapes can be well reproduced by assigning the non-Voigt effects to speed-dependent effects rather than to collisional narrowing. These facts justify the choice of a reduced model against a generalized higher-order (HTP) parameterization.

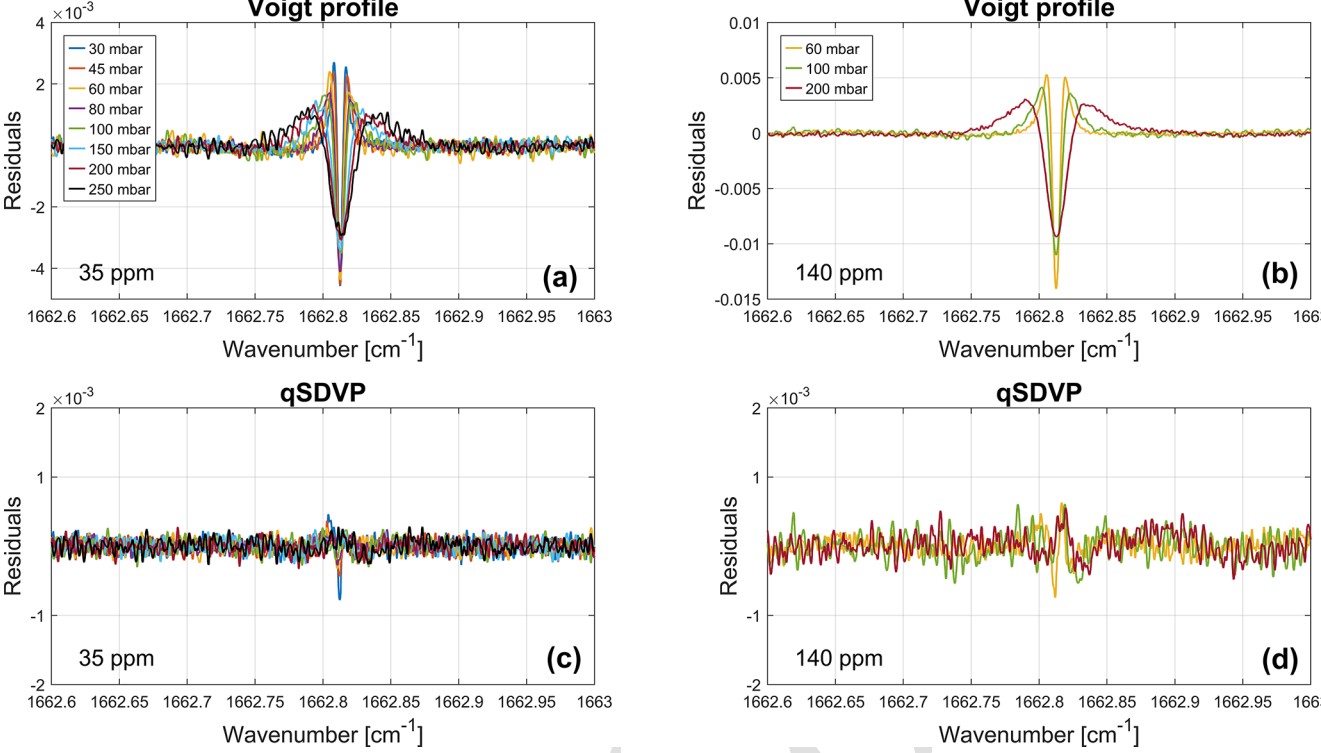

**Figure 6.** Fit residuals obtained using a Voigt profile (VP; **a–b**) and quadratic speed-dependent Voigt profile (qSDVP; **c–d**) line shape model at 35 ppm $H_2O$ (**a, c**) and 140 ppm $H_2O$ (**b, d**), color-coded with pressure. The VP fit is performed using molecular parameters from the HITRAN2020 database, while the qSDVP fitting uses the $\Gamma_0$ and $\Gamma_2$ values determined from the MSF approach.

## 4 Results

### 4.1 Precision and long-term stability

The precision and long-term stability of ALBATROSS are assessed using 30 min of data recorded at 1 s resolution, taken at the end of each measurement interval (see Fig. 2b). The Allan–Werle deviation technique (Allan, 1966; Werle et al., 1993) was used in particular to determine the measurement precision as a function of integration time and to distinguish between instrumental drifts and random noise (Werle et al., 1993). The analysis was repeated using both the VP and the qSDVP retrievals, revealing no impact on the measurement precision by the choice of line shape model. The results shown in the following were obtained with the qSDVP model and the parameters determined as discussed above.

Figure 7 shows frequency of occurrence histograms (left) and Allan–Werle deviation plots (right) for a selection of $H_2O$ amount fractions and pressure levels. The frequency of occurrence distributions, color-coded with pressure, corresponds to 2.5, 10, and 35 ppm $H_2O$ (panels a–c). Each distribution is calculated in 40 bins of 5 ppb width, centered around the mean $H_2O$ amount fraction. Allan–Werle deviations as a function of integration time are shown for all pressures at 2.5 ppm $H_2O$ (panel d) and for all $H_2O$ set points at 150 mbar (panel e). The theoretical line corresponding

to white-noise behavior ($\sim \tau^{-1/2}$ where $\tau$ is the integration time) is also indicated in panels (d)–(e) as a reference.

The frequency of occurrence distributions is generally very narrow (within $\pm 30$ ppb) in all conditions (Fig. 7a–b), with a few exceptions at low pressures and high water amount fractions (Fig. 7c). The standard deviations ($\sigma_{1\,s}$) of the distributions shown in Fig. 7a–c vary between 6 and 9 ppb (corresponding to 0.23 %–0.33 % of the reference value) at 2.5 ppm $H_2O$, 7–13 ppb (0.07 %–0.13 %) at 10 ppm $H_2O$, and 9–30 ppb (0.03 %–0.08 %) at 35 ppm $H_2O$. On average over all pressures, $\sigma_{1\,s}$ lies between 0.04 % (14 ppb) at 35 ppm $H_2O$ and 0.25 % (7 ppb) at 2.5 ppm $H_2O$.

The Allan–Werle deviation plots indicate a stable operation of the spectrometer up to 100 s, after which drifts start to dominate the measurements. This can be due to mechanical and optical instabilities of the instrument but also due to flow rate fluctuations in the MFCs. The Allan deviation minimum is always reached at integration times between 50 and 100 s, corresponding to a precision between 0.5 and 5 ppb (i.e., 0.02 %–0.5 % of the reference value). Therefore, an integration time of 50 s was selected for the accuracy assessment and for the determination of the qSDVP parameters. Investigating the time series of the zero air measurements over longer timescales indicated that the spectrometer maintains a stable operation over a few hours, at least. The Allan–Werle

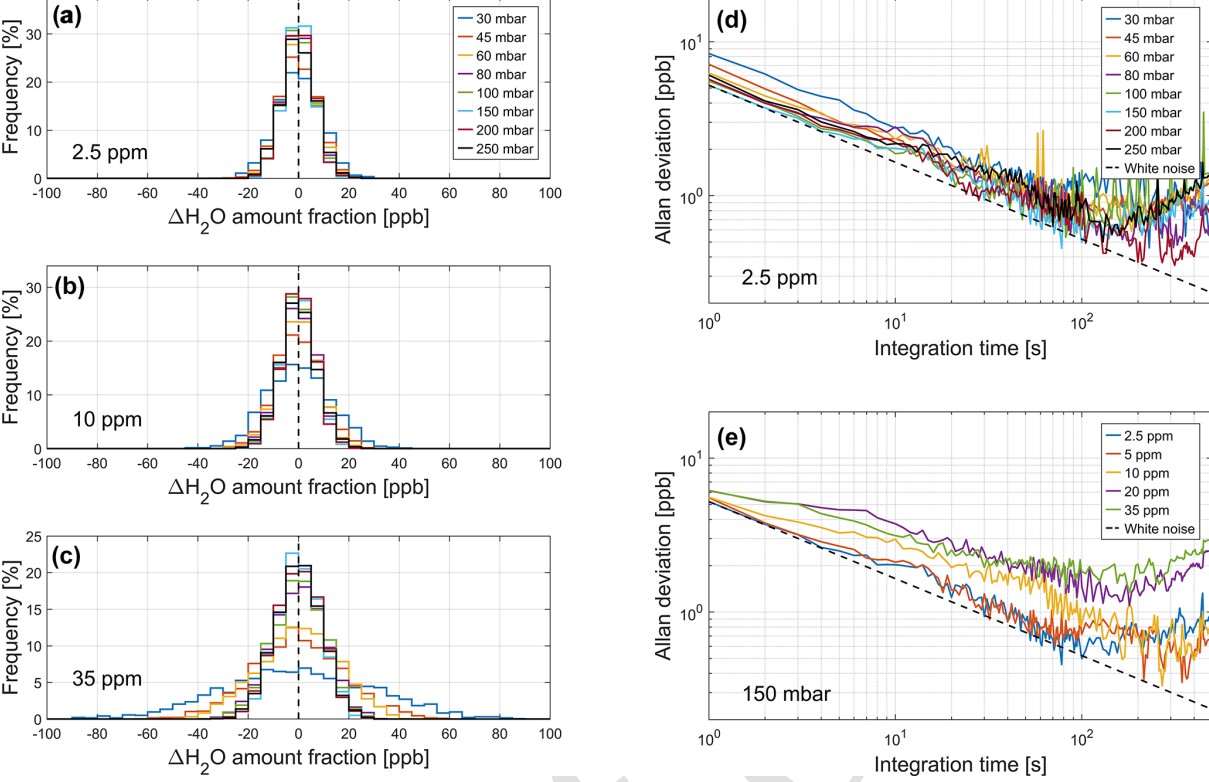

**Figure 7.** Precision and long-term stability assessment. **(a–c)** Frequency of occurrence distributions calculated over 30 min of measurements at 1 s resolution for 2.5 ppm $H_2O$ **(a)**, 10 ppm **(b)**, and 35 ppm **(c)**, color-coded with pressure. Each distribution is calculated using 40 bins of 5 ppb width, centered on the mean $H_2O$ amount fraction measured at the corresponding set point and pressure. **(d–e)** Allan–Werle deviation plots for all pressures at 2.5 ppm $H_2O$ **(d)** and for all $H_2O$ set points at pressure level 150 mbar **(e)**. The white-noise behavior ($\sim \tau^{-1/2}$, where $\tau$ is the integration time) is indicated in panels **(d)**–**(e)** as a reference.

deviation minimum in this case stayed at a constant value of 0.5 ppb $H_2O$ after about 2 min averaging (see Fig. S3).

## 4.2 Accuracy

The accuracy of ALBATROSS is evaluated by comparing the spectroscopically retrieved $H_2O$ amount fractions with the SI-traceable reference $H_2O$ values generated by the dynamic–gravimetric method (as listed in Table 1), at each pressure level. To quantify the improvement in accuracy obtained by the qSDVP parameterization, the results are also compared to those obtained using the VP line shape model and HITRAN2020 molecular parameters.

Figure 8 shows the relative difference in $H_2O$ amount fractions between the ALBATROSS retrievals (integrated over 50 s) and the reference values ($\Delta H_2O$) as a function of pressure and color-coded with the $H_2O$ set point. The error bars correspond to $1\sigma$ standard deviation. The relative uncertainty in the $H_2O$ reference levels generated by the permeator ($\pm 1.5\%$) is indicated by the grey shaded area.

The results show that all measurements retrieved using the qSDVP line shape model are found within the $\pm 1.5\%$ uncertainty range of the permeator and are hence in excellent

agreement with the SI-traceable reference values. In contrast, the VP retrievals (with $\Gamma_0$ and $S_{ij}$ from HITRAN2020) systematically overestimate the amount fractions by $1.5\%$–$5\%$ compared to the reference, with a bias increasing with pressure.

Using qSDVP, the largest relative deviation from the reference occurs at 2.5 ppm $H_2O$ and 45 mbar ($+1.05\%$, corresponding to $+27$ ppb $H_2O$), while the largest absolute deviation is found at 35 ppm $H_2O$ and 100 mbar ($+0.79\%$, i.e., $+273$ ppb $H_2O$). On average over all pressures, $\Delta H_2O$ varies between TS1 $+0.01\%$ (i.e., $+0.1$ ppm) at 10 ppm $H_2O$ and $+0.22\%$ ($+5.5$ ppm) at 35 ppm $H_2O$. The pressure-averaged $\Delta H_2O$ values and their corresponding standard deviations are summarized in Table 3. This excellent agreement reflects the high quality of the molecular parameters listed in the HITRAN database and their well-confined uncertainties, at least for the $H_2O$ transition selected for our study. Furthermore, it also demonstrates that beyond-Voigt line profile models have the capability to accurately describe the observed shape of the absorption line, opening the path to a highly accurate quantification of the observed data.

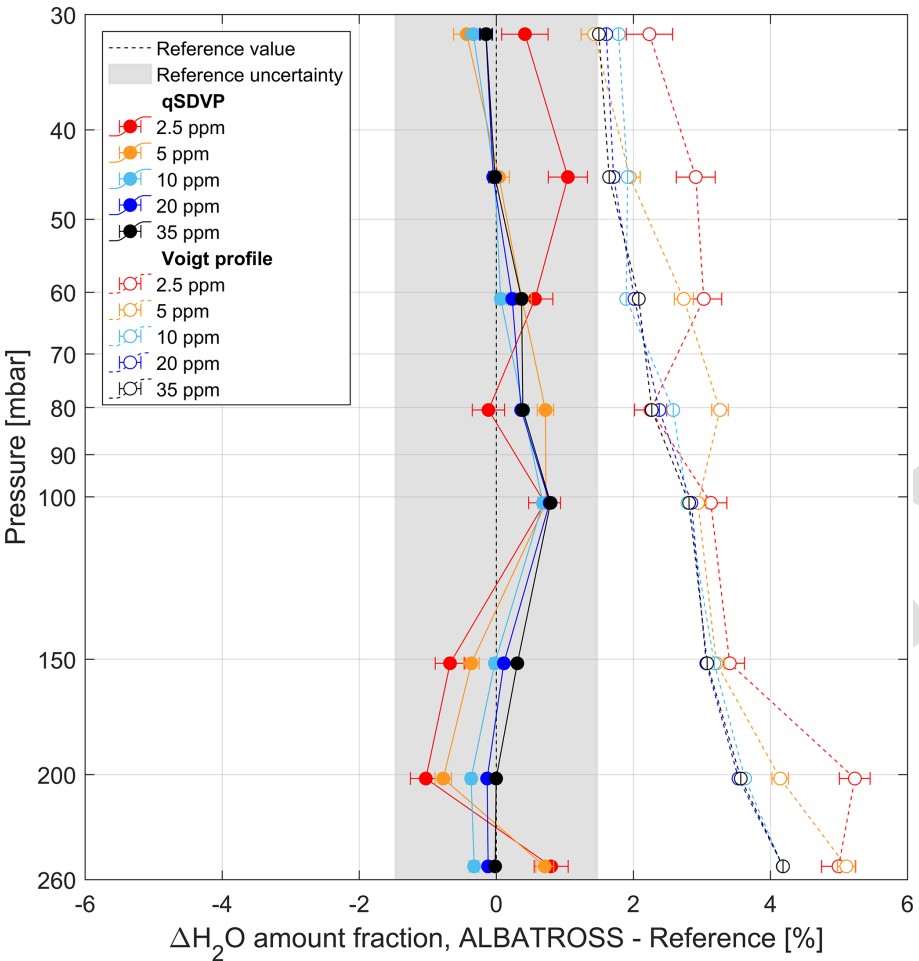

**Figure 8.** Accuracy assessment. The relative difference in $H_2O$ amount fractions between the spectroscopic retrievals and the reference values ($\Delta H_2O$) as a function of pressure, color-coded with $H_2O$ amount fractions. Results obtained using the qSDVP line shape model are shown as filled circles and solid lines, while Voigt profile (VP) results are shown as open circles and dashed lines. The error bars for each data point correspond to the $1\sigma$ standard deviation. The expanded relative uncertainty in the $H_2O$ reference levels generated by the dynamic–gravimetric method ($\pm 1.5\%$) is indicated by the grey shaded area.

**Table 3.** Accuracy and precision of the ALBATROSS measurements performed during the validation. Absolute and relative deviations are expressed as $\Delta H_2O \pm 1\sigma$, where $\Delta H_2O$ is the $H_2O$ amount fraction difference between the ALBATROSS retrievals and the SI-traceable reference levels and $\sigma$ the standard deviation, both averaged over all the considered pressure levels (30–250 mbar).

| Accuracy and precision | | |
| --- | --- | --- |
| Set point (ppm) | Absolute deviation (ppb) | Relative deviation (%) |
| 2.5 | $+5.5 \pm 7$ | $+0.22 \pm 0.25$ |
| 5 | $+6.1 \pm 7$ | $+0.12 \pm 0.14$ |
| 10 | $+0.1 \pm 8$ | $+0.01 \pm 0.08$ |
| 20 | $+26 \pm 11$ | $+0.13 \pm 0.05$ |
| 35 | $+72 \pm 14$ | $+0.21 \pm 0.04$ |

### 4.3 Linearity (extended-range validation)

The aim of the extended-range validation was to characterize the linearity of ALBATROSS beyond the upper limit of 35 ppm $H_2O$ delivered by the dynamic–gravimetric method. For this, the custom-made secondary reference gas (not SI traceable), prepared as described in Sect. 2.3, was diluted to four different $H_2O$ amount fraction levels using two mass flow controllers (MFCs 4–5 in Fig. 1). The linearity of the spectrometer is assessed by comparing the spectroscopically retrieved $H_2O$ amount fractions with the calculated amount fractions based on the dilution ratio and the $H_2O$ content of the undiluted reference gas.

The MFCs (Axetris, Switzerland) were calibrated at METAS using an SI-traceable primary standard with an accuracy of 0.2 % in the volumetric flow rate. The dilution ratio is defined as the ratio of the flow rate of the secondary

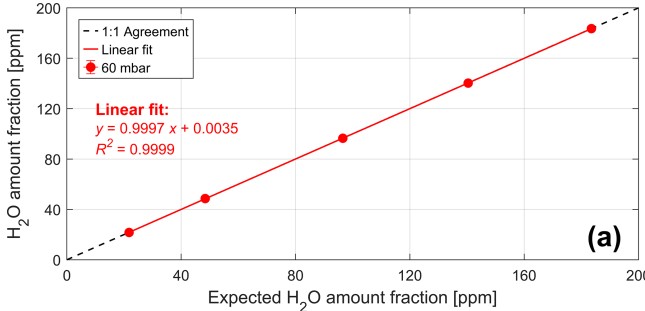

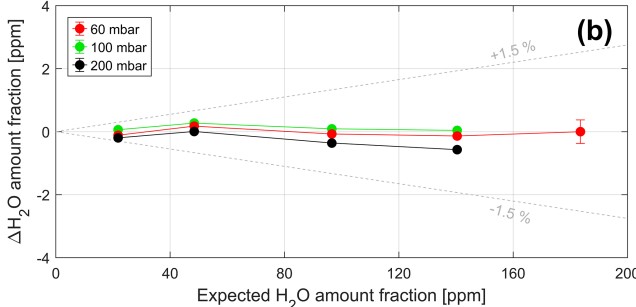

**Figure 9.** Linearity assessment over the extended range. Correlation plot of the $H_2O$ amount fractions retrieved by ALBATROSS ($y$ axis) versus the expected $H_2O$ amount fractions based on the dilution ratio ($x$ axis) at 60 mbar **(a)** and their differences ($\Delta H_2O$) as a function of the expected amount fractions at 60, 100, and 200 mbar **(b)**. The dashed grey lines in panel **(b)** indicate a relative deviation of $\Delta H_2O = \pm 1.5\%$.

reference gas to the total flow rate (kept constant) of the secondary reference and dilution gas. The $H_2O$ amount fraction of the undiluted secondary reference gas, namely $183.54 \pm 0.06$ ppm, was determined by ALBATROSS at 60 mbar using the qSDVP line shape model and integration time of 50 s. The calculated $H_2O$ amount fractions and their corresponding uncertainties obtained for each dilution step are listed in Table 4.

Figure 9 shows a scatterplot of the $H_2O$ amount fractions retrieved by ALBATROSS (using qSDVP and integrated over 50 s) at 60 mbar versus the expected $H_2O$ amount fractions (panel a) and their difference ($\Delta H_2O$) at 60, 100, and 200 mbar (panel b). The results of a linear fit (slope, intercept, and determination coefficient $R^2$) between the measured and the expected $H_2O$ amount fractions at 60 mbar are also displayed in panel (a). As the $H_2O$ content of the undiluted cylinder was only determined once, we estimate its uncertainty based on the repeatability of the measurement performed at 140 ppm (i.e., 310 ppb $H_2O$) and the precision of the single measurement at 180 ppm (60 ppb $H_2O$). This results in a total uncertainty of $\pm 370$ ppb $H_2O$, i.e., roughly $\pm 0.2\%$ (as shown by the error bars in Fig. 9b).

All spectroscopically retrieved $H_2O$ amount fractions are found to be in very good agreement with the expected values (within $\pm 0.6\%$) and with an excellent correlation over the entire investigated range (Fig. 9). On average over all pressures, $\Delta H_2O$ varies between $-0.22$ ppm (i.e., $-0.16\%$) at 140 ppm $H_2O$ and $+0.15$ ppm ($+0.31\%$) at 48 ppm $H_2O$. The largest relative deviation was found at 22 ppm $H_2O$ and 200 mbar ($-0.92\%$, i.e., $-0.20$ ppm). The $1\sigma$ standard deviation varies between 13 ppb ($0.06\%$) at 22 ppm $H_2O$ and 30 ppb ($0.02\%$) at 140 ppm $H_2O$ (see Table 4).

Along with the results obtained with the SI-traceable reference gases in the range of 2.5–35 ppm $H_2O$, the extended-range validation demonstrates the outstanding performances of ALBATROSS in conditions that fully cover the expected variability in $H_2O$ in the UTLS (see Fig. 2a). Although these assessments were performed in a well-controlled laboratory environment using a closed-path configuration, we expect that the performance of the instrument will remain the same during flight conditions too. The demonstration of this behavior is beyond the scope of this study; however, some related aspects (the effects of turbulence, thermomechanical deformation, etc.) were described in our earlier studies (e.g., Tuzson et al., 2020; Graf, 2020; Graf et al., 2021). We found that the highly reduced optical complexity of ALBATROSS exhibits an excellent robustness against environmental impacts, and its performance did not deteriorate.

This performance in terms of accuracy and precision makes ALBATROSS a particularly attractive tool for investigating the upper-troposphere region. This can be of interest to cirrus cloud microphysical modeling (e.g., Luo et al., 2003; Reinares Martínes et al., 2020) and in-cloud supersaturation studies (e.g., Krämer et al., 2009; Dekoutsidis et al., 2023), requiring accurate $H_2O$ measurements at high vertical/temporal resolution as input parameters.

## 5  Conclusions

We have presented a detailed laboratory assessment of ALBATROSS, a newly developed quantum-cascade laser absorption spectrometer for balloon-borne measurements of $H_2O$ in the upper troposphere–lower stratosphere (UTLS). The validation was performed using SI-traceable reference gases generated by a dynamic–gravimetric permeation method, capable of delivering $H_2O$ amount fractions as low as 2.5 ppm in synthetic air, with an uncertainty smaller than $\pm 1.5\%$. The accuracy and precision of ALBATROSS were evaluated in a wide range of pressure (30–250 mbar) and $H_2O$ amount fractions (2.5–35 ppm), representative of the atmospheric variability in $H_2O$ in the UTLS. In addition, the linearity of ALBATROSS was verified in up to 180 ppm $H_2O$ using a custom-made secondary reference gas mixture.

We found that the quadratic-speed-dependent Voigt profile (qSDVP) gives a quality of the fit that is commensurate with our spectral signal-to-noise ratio and accurately reproduces the measured line shapes without any systematic bias over the entire pressure range. The molecular parameters required by this parameterization ($\Gamma_0$ and $\Gamma_2$) for the $H_2O$ line inves-

**Table 4.** Summary of the extended-range validation measurements. Dilution ratios and calculated $H_2O$ amount fractions are expressed as expected values $\pm$ the uncertainty associated with the volumetric flow rate measurements by the mass flow controllers. Measured $H_2O$ amount fractions are expressed as mean values $\pm 1\sigma$ the standard deviation, averaged over the three pressure levels considered here (60, 100, and 200 mbar). Absolute and relative deviations are the differences between measured and calculated $H_2O$ amount fractions. Note that the undiluted $H_2O$ amount fraction of the secondary reference gas (i.e., dilution ratio $= 1$) was measured by ALBATROSS at 60 mbar and is used to calculate the expected $H_2O$ amount fractions at all levels.

| Linearity (extended-range validation) | | | | |
| --- | --- | --- | --- | --- |
| Dilution ratio | Calculated $H_2O$ amount fraction (ppm) | Measured $H_2O$ amount fraction (ppm) | Absolute deviation (ppm) | Relative deviation (%) |
| $0.1187 \pm 0.0004$ | $21.78 \pm 0.07$ | $21.69 \pm 0.01$ | $-0.08$ | $-0.38$ |
| $0.2637 \pm 0.0008$ | $48.41 \pm 0.15$ | $48.56 \pm 0.02$ | $+0.15$ | $+0.31$ |
| $0.5261 \pm 0.001$ | $96.56 \pm 0.18$ | $96.45 \pm 0.02$ | $-0.11$ | $-0.10$ |
| $0.7650 \pm 0.0007$ | $140.41 \pm 0.13$ | $140.19 \pm 0.03$ | $-0.22$ | $-0.16$ |
| 1 | – | $183.54 \pm 0.06$ | – | – |

tigated here were determined experimentally using a multi-spectrum fitting (MSF) approach over multiple pressure conditions. Furthermore, we demonstrated that the implementation of the qSDVP line shape model, using these empirically determined broadening parameters (given in Table 2), improves the accuracy by up to 5 % compared to the VP model.

The measurements show that ALBATROSS achieves an accuracy better than $\pm 1.5$ % with respect to the SI-traceable reference at all investigated pressures and $H_2O$ amount fractions. The measurement precision at 1 s resolution was found to be better than 30 ppb (i.e., 0.1 % at 35 ppm $H_2O$) in all conditions and as low as 5 ppb (0.02 % at 35 ppm $H_2O$) upon integrating the data over 50 s, corresponding to the Allan deviation minimum. These results are particularly remarkable considering the technical challenges of maintaining a stable $H_2O$ amount fraction level in the low parts per million range in a laboratory setting, due to the strong surface adsorption/desorption properties of $H_2O$. Furthermore, the comparison between the normalization methods, i.e., empty cell and polynomial baseline (shown in the Supplement), demonstrates the applicability of these results to the analysis of real atmospheric (i.e., open-path) data as well.

The performance achieved by ALBATROSS is unprecedented for a lightweight balloon-borne hygrometer and thus demonstrates the exceptional potential of mid-IR laser absorption spectroscopy as an alternative reference method to cryogenic frost-point hygrometry (CFH) for in situ measurements of $H_2O$ in the UTLS. This is particularly relevant considering the ongoing phasing out of the cooling agent required by CFH (fluoroform, HFC-23), which drives the need for an alternative solution to maintain the monitoring of UTLS $H_2O$ in global, long-term monitoring networks, such as the GRUAN.

Currently, ALBATROSS is participating in the AquaVIT-4 International Intercomparison of Atmospheric Hygrometers, held at the AIDA cloud simulation chamber (Karlsruhe Institute of Technology, Germany). This will allow us to characterize the performance of the spectrometer under a new set of challenging UTLS-relevant conditions, also in terms of temperature, and to validate its accuracy against a set of well-established hygrometers (both laboratory based and airborne). Finally, new intercomparison test flights between CFH and ALBATROSS are foreseen to further validate the performances achieved in the laboratory in a real atmospheric environment.

*Data availability.* The data are available from the authors upon request.

*Supplement.* The supplement related to this article is available online at: https://doi.org/10.5194/amt-16-1-2023-supplement.

*Author contributions.* SB, MG, and TB performed the measurements under the supervision of BT and CP. SB performed the data analysis with support from MG, II, and BT. BT and LE supervised the project. SB and BT wrote the paper with contributions from all authors.

*Competing interests.* The contact author has declared that none of the authors has any competing interests.

*Acknowledgements.* The ALBATROSS (balloon-borne laser spectrometer for UTLS water research) project was funded by the Federal Office of Meteorology and Climatology (MeteoSwiss), in the framework of GCOS (Global Climate Observing System), Switzerland. The authors thank Martin Vollmer (Empa) for support in the

generation of the secondary reference gas mixture. Philipp Scheidegger, André Kupferschmid, and Herbert Looser are acknowledged for their continuous HW and SW support. We are grateful to Alan Fried, Markus Miltner, Daniele Romanini, and the anonymous reviewer for their constructive comments and valuable suggestions which helped us to improve this paper.

*Review statement.* This paper was edited by Thomas Röckmann and reviewed by Alan Fried, Daniele Romanini, Markus Miltner, and one anonymous referee.

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

**Remarks from the typesetter**

TS1    Please note that the corrections of "numbers" are not language changes. If you still insist on changing these values in this caption and Table 3, the editor has to approve these changes. Please give an explanation of why these values need to be changed.