# Peer review of "SI-traceable validation of a laser spectrometer for balloon-borne measurements of water vapor in the upper atmosphere"

_Atmospheric Measurement Techniques, 2023_

## Referee Comment (RC1)

Review of AMT Paper – May 20, 2023

" SI-traceable validation of a balloon-borne spectrometer for water vapor measurements in the upper atmosphere"
By

Simone Brunamonti, Manuel Graf, Tobias Bühlmann, Céline Pascale, Ivan Ilak, Lukas Emmenegger and Béla Tuzson

This is an impressive study which describes a newly developed QCL-based absorption spectrometer for highly accurate balloon-borne water vapor measurements in the UTLS. The instrument was validated in the laboratory employing SI-traceable reference gas standards from: a dynamically diluted gravimetric permeation system; and from gravimetrically prepared compressed gas water vapor mixtures. Various absolute water vapor mixing ratios from 2.5 ppm up to 180 ppm were generated and measured over the range of temperature and pressure conditions representative of the UTLS.

Using these standards, improved spectroscopic broadening and pressure shift parameters were determined employing a quadratic-speed dependent Voigt fitting profile, which accurately reproduced the measured line shapes without systematic residuals over the entire pressure range. The resulting fitting approach and empirically determined parameters improved the resulting accuracy over the more traditional Voigt line profile using HITRAN 2020 parameters. The authors clearly demonstrated that their balloon-borne instrument is capable of retrieving UTLS water vapor mixing ratios with a calibration-free instrument with an absolute accuracy to better than $\pm$ 1.5 % with respect to the SI-standards.

The results of this study and the future deployment of this instrument will represent a significant advancement in UTLS measurements of the most important Greenhouse gas. This paper is Excellent in its: Scientific Significance, Scientific Quality, and the Quality of Presentation. This reviewer highly recommends publication with only minor revisions, as detailed below, for improved clarity.

1. In the Abstract, it would be desirable to define the acronym ALBATROSS

2. Line 18, page 2: It would be useful to the reader to indicate right up front to write: "The aim of this work is to validate the accuracy and precision of a newly developed **open path** mid-IR quantum-cascade …..", even though this is indicated in the next section.

3. Page 3, Starting on Line 13 in the discussion of "Rapid spectral sweeping …", although this approach is discussed in Graf et al. (2021), it would be useful for the reader to add another sentence right after "….Liu et al., 2018) further explaining this approach here. For example, you could add " In this approach the laser driving current is applied in pulses, followed by a moment of complete shutdown of the laser to re-establish its initial temperature".

4. Regarding Fig. 1, I am a little confused by the conceptual diagram of the solenoid valve following the diluted permeation source. I would expect the common port on the MFC3 end and not on the Zero air input end. Also, I was expecting a multi-port valve whereby the diluted permeation flow is directed out to a vent port without flow disruption when the Zero air is switched in. I realize this is a conceptual diagram, but it would be important for clarity to represent this more accurately.

   Also, for completeness the authors should mention whether or not they observe any hysteresis in the time response passing through MFC3 into the MP cell upon switching in and out the diluted permeation flow and the Zero air. Is the 3 hour equilibrium time dictated by hysteresis in this MFC, by the MP cell, by the connecting lines, or all of the above? Also, a brief discussion of the actual liquid water source used in the permeation device and its purity is warranted

5. Table 1 headings and descriptor at the bottom, needs further clarification/modification. The labels "Nominal value" and "Actual value" need to be defined more clearly in the descriptor below. These terms refer to the generated water vapor mixing ratios from the dynamic-gravimetric method and the spectroscopic measured values, respectively. The descriptor "expanded measurement uncertainty" is confusing here, as I believe you are combining two terms: the expansion in the number of significant figures in the determined spectroscopic values from the nominal values expected from the generation system and the combined uncertainty from all the parameters employed in the spectroscopic measurement? Is this correct. Are these uncertainty limits at the 1σ levels? Do the nominal values from the standards generation system have additional significant figures than those shown in Table 1? I ask this because aren't you ultimately relying on agreement between your qSDVP fitting procedure and the standards generation system output (as implied on page 10, line 3)? If your absolute water mixing ratios from the standards generation system does not have more significant figures than those indicated in Table 1, aren't you reporting too many significant figures in your expanded actual values? Perhaps some clarification here?

6. The comparison between the polynomial baseline fitting and the empty cell approach is impressive and is important since only the former can be used in real atmospheric measurements.

---

## Author Comment (AC1)

We would like to thank the Referee for the thoughtful and constructive comments that helped us to improve our manuscript. We addressed each comment individually and made revisions in response to their suggestions, as detailed below. Our replies to the Referee comments are highlighted in blue. Modifications to the manuscript text are indicated in *italic*.

**Referee #1 (Alan Fried)**

This is an impressive study which describes a newly developed QCL-based absorption spectrometer for highly accurate balloon-borne water vapor measurements in the UTLS. The instrument was validated in the laboratory employing SI-traceable reference gas standards from: a dynamically diluted gravimetric permeation system; and from gravimetrically prepared compressed gas water vapor mixtures. Various absolute water vapor mixing ratios from 2.5 ppm up to 180 ppm were generated and measured over the range of temperature and pressure conditions representative of the UTLS.

Using these standards, improved spectroscopic broadening and pressure shift parameters were determined employing a quadratic-speed dependent Voigt fitting profile, which accurately reproduced the measured line shapes without systematic residuals over the entire pressure range. The resulting fitting approach and empirically determined parameters improved the resulting accuracy over the more traditional Voigt line profile using HITRAN 2020 parameters. The authors clearly demonstrated that their balloon-borne instrument is capable of retrieving UTLS water vapor mixing ratios with a calibration-free instrument with an absolute accuracy to better than ±1.5 % with respect to the SI-standards.

The results of this study and the future deployment of this instrument will represent a significant advancement in UTLS measurements of the most important Greenhouse gas. This paper is Excellent in its: Scientific Significance, Scientific Quality, and the Quality of Presentation. This reviewer highly recommends publication with only minor revisions, as detailed below, for improved clarity.

We are very grateful for your nice words and supportive feedback.

1. In the Abstract, it would be desirable to define the acronym ALBATROSS

The acronym ALBATROSS is a somewhat freely chosen name resulting from a random pick of letters of a descriptive phrase: "Balloon-borne laser spectrometer for UTLS water research". To avoid repetitive phrases in our abstract, we decided to leave the definition in the Acknowledgement section.

2. Line 18, page 2: It would be useful to the reader to indicate right up front to write: "The aim of this work is to validate the accuracy and precision of a newly developed **open path** mid-IR quantum-cascade …..", even though this is indicated in the next section.

We agree, implemented as suggested.

3. Page 3, Starting on Line 13 in the discussion of "Rapid spectral sweeping …", although this approach is discussed in Graf et al. (2021), it would be useful for the reader to add another sentence right after "….Liu et al., 2018) further explaining this approach here. For example, you could add "

In this approach the laser driving current is applied in pulses, followed by a moment of complete shutdown of the laser to re-establish its initial temperature".

We have slightly modified the respective paragraph to include more details about the laser driving as suggested:

*"Rapid spectral sweeping of the QCL is achieved by periodic modulation of the laser driving current, following the intermittent continuous wave (ICW) modulation approach (Fischer et al., 2014). In this approach, the laser driving current is applied in pulses (typically 100 µs long), followed by a short period of complete shutdown of the QCL. The ICW driving is obtained using custom developed analogue electronics (Liu et al., 2018)."*

4. Regarding Fig. 1, I am a little confused by the conceptual diagram of the solenoid valve following the diluted permeation source. I would expect the common port on the MFC3 end and not on the Zero air input end. Also, I was expecting a multi-port valve whereby the diluted permeation flow is directed out to a vent port without flow disruption when the Zero air is switched in. I realize this is a conceptual diagram, but it would be important for clarity to represent this more accurately.

Indeed, the common port of the solenoid valve is connected to MFC3, in order to switch between the diluted permeation source and zero air. The orientation of the solenoid valve is now indicated by the labels "COM" (MFC 3), "NO" (normally open - zero air) and "NC" (normally closed - permeation source) in the schematics. Also, there was an overflow between the solenoid valve and MFC3, which was not indicated in the previous schematics. This is now included in the revised version of Figure 1.

Also, for completeness the authors should mention whether or not they observe any hysteresis in the time response passing through MFC3 into the MP cell upon switching in and out the diluted permeation flow and the Zero air. Is the 3 hour equilibrium time dictated by hysteresis in this MFC, by the MP cell, by the connecting lines, or all of the above?

The response time of the instrument after switching between the diluted permeation flow and zero air varied in function of two factors: the gas pressure and the water vapor content. A lower gas pressure resulted in systematically longer response times and higher zero levels that suggests enhanced desorption effects of $H_2O$ from surfaces. Furthermore, an elevated water vapor content involved a longer response time, a clear indication for memory (surface) effects. The complex interplay of these two mechanisms hinders us to make any quantitative statement of such hysteresis effects, also because of the lack of repeated measurements at the same pressure conditions.

Nevertheless, the contribution of the MFC to the response time of the instrument was evaluated at the end of the campaign, by connecting the synthetic air source directly to the multipass cell (i.e., bypassing MFC3 and the solenoid valve). This showed no significant variation in the zero level reached by the instrument, indicating that the role played by the MFC is negligible. Hence, the effects discussed above are mainly due to the multipass cell and the sampling line, although, their relative contributions were not assessed in details.

We added a more detailed description of the observed effects in Section 2.4 of the revised manuscript:

*"This behavior is mainly due to the strong surface adsorption/desorption properties of $H_2O$, causing*

*a memory effect to the system. Throughout the validation experiments, we observed a tendency towards slightly elevated zero levels whenever the gas pressure in the sampling line was lowered. Furthermore, the response time of the instrument showed a clear correlation with the humidity content of the measured gas. While these effects may, if not properly taken into account, e.g. by the empty-cell spectrum normalization, affect the accuracy of the measurements, they are largely absent during flight conditions, where the instrument is operated in open-path configuration. In this case, there is no sampling line, the gas flow is much larger, and the surfaces of the SC-MPC are drastically reduced, as the lids are removed and the gas-surface interaction is limited to the narrow inner circumference of the cell."*

Also, a brief discussion of the actual liquid water source used in the permeation device and its purity is warranted

The water within the permeation device was ultrapure water. This information is now included in Section 2.2 of the revised manuscript.

*"The permeator […] was filled with ultrapure water (resistivity 18.2 MΩ cm at 25 °C, corresponding to a purity > 99.999 %)."*

5. Table 1 headings and descriptor at the bottom, needs further clarification/modification. The labels "Nominal value" and "Actual value" need to be defined more clearly in the descriptor below. These terms refer to the generated water vapor mixing ratios from the dynamic-gravimetric method and the spectroscopic measured values, respectively. The descriptor "expanded measurement uncertainty" is confusing here, as I believe you are combining two terms: the expansion in the number of significant figures in the determined spectroscopic values from the nominal values expected from the generation system and the combined uncertainty from all the parameters employed in the spectroscopic measurement? Is this correct. Are these uncertainty limits at the $1\sigma$ levels? Do the nominal values from the standards generation system have additional significant figures than those shown in Table 1? I ask this because aren't you ultimately relying on agreement between your qSDVP fitting procedure and the standards generation system output (as implied on page 10, line 3)? If your absolute water mixing ratios from the standards generation system does not have more significant figures than those indicated in Table 1, aren't you reporting too many significant figures in your expanded actual values? Perhaps some clarification here?

The label "Nominal value" in Table 1 does not refer to the actual $H_2O$ amount fractions generated by the dynamic-gravimetric method, but rather to the *target* $H_2O$ amount fractions that we aimed to generate by this method. Conversely, the "Actual value" refers to the *actual* generated $H_2O$ amount fractions (and their expanded uncertainties), determined upon calibration of the permeator and the dynamic dilution unit before and after the measurement campaign. The spectroscopically measured $H_2O$ amount fractions are not given in Table 1, but only in Table 3 (expressed as differences with respect to the "Actual values" generated by the dynamic-gravimetric method). Hence, the number of reported significant figures is consistent with the accuracy of the reference gas generation method ($< 1.5$ % at all conditions).

For the sake of clarity, we have now replaced "Nominal value" with "Target value" in Table 1, and keep "Actual value" to indicate the $H_2O$ amount fractions generated by the permeator. The same applies to the pressure values given in the same table.

6.    The comparison between the polynomial baseline fitting and the empty cell approach is impressive and is important since only the former can be used in real atmospheric measurements.

Good remark. We added a sentence in the Conclusions to highlight this point.

*"Furthermore, the comparison between the normalization methods, i.e., empty-cell and polynomial-baseline (shown in the Supplementary material) demonstrates the applicability of these results to the analysis of real atmospheric (open-path) data as well."*

---

## Author Comment (AC2)

We would like to thank the Referee for the thoughtful and constructive comments that helped us to improve our manuscript. We addressed each comment individually and made revisions in response to their suggestions, as detailed below. Our replies to the Referee comments are highlighted in blue. Modifications to the manuscript text are indicated in *italic*.

**Referee #2 (Markus Miltner)**

**General comments**

This paper is of interest for the AMT audience for several reasons: 1) It demonstrates that laser absorption spectrometers are a promising alternative to cryogenic frost-point hygrometers under the particularly difficult measurement conditions encountered in the upper atmosphere (variable pressure, low and variable water concentration). 2) It gives a blueprint of a laboratory validation for such measurement devices. 3) It shows the importance of choosing a more advanced (compared to the Voigt profile) line shape model (here the qSDVP) and demonstrates how to obtain the necessary parameters, which are not contained in the HITRAN database.

The experiment is well-designed and the underlying measurements and data analysis are presented in detail, allowing fellow scientists to reproduce the measurements if they wish.

The paper is well written: It is easy to follow the authors through the chapters, thanks to the clear structure, the precise language and the supporting figures.

**Specific comments**

1 Introduction, p2, l25; State that only pressure broadening parameters are assessed and the line strength is not (since only pressure was varied, while temperature was held constant). This was not immediately clear to me when first reading the paper.

We agree that the description was not entirely complete and added a clarifying sentence to the Introduction as suggested:

*"[…] such as the line strength, its temperature dependence, and the pressure broadening parameters. Especially, a detailed knowledge of the latter is a prerequisite for an accurate spectral retrieval. Our focus was on the broadening effects, while for the line strength and its temperature dependency we take the values from the HITRAN 2020 database."*

2.3 Gas handling system, p4, l13; Flow rates not consistent with what is stated in Figure 1 (0.05 to 3 slpm vs 0.05 to 4.5 slpm)

Text revised: 0.05-4.5 slpm is the correct range.

Figure 1; Should there not be a vent somewhere between the mixing solenoid valve and MFC 3? If not, where does the excessive flow go? Please clarify this.

Indeed, an overflow valve was present between the solenoid valve and MFC 3, which was neglected in the previous schematics. It is now included in the revised version of Figure 1.

p4, l19; how was the temperature controlled?

The experiments were performed in an air conditioned laboratory at METAS with set temperature of 20 °C. The spectrometer was operated inside a plexiglass chamber (at ambient pressure) to further suppress any sudden temperature fluctuations from outside. The temperature inside the PMMA chamber (next to the multipass cell) was continuously monitored by the Vaisala HMP110 sensor and found to be stable at 23.5°C with a standard deviation of 0.03 °C. This information was now included in Section 2.3 of the revised manuscript, and the PMMA chamber is now also indicated on the schematics in Figure 1.

*"All measurements were performed in an air conditioned laboratory and the spectrometer was operated inside a custom-made PMMA plastic chamber (thickness 15 mm) to further suppress any sudden temperature or humidity fluctuations. The temperature was monitored in the vicinity of the multipass cell (inside the PMMA chamber)."*

P4, l25; it would be interesting for fellow scientists trying to reproduce the experiments to know how the gas cylinder used for the secondary reference gas mixture was "conditioned" and what kind of synthetic air (upper boundary of water content?) was used to prefill and pressurize the bottle.

The only purpose of this experiment was to demonstrate that the QCLAS maintains its linearity over an extended range of water content, beyond the levels obtained from the permeation method. We used a regular steel cylinder (previously containing synthetic air) and after its evacuation, we applied the procedure as described in the manuscript. The conditioning was achieved by repeated purging (with synthetic air) and evacuation steps. The synthetic air used to fill the gas cylinder contained $< 5$ ppm $H_2O$ (according to the specification sheets, Messer, Switzerland). The water spiking was immediately followed by the filling process. After filling, the gas bottle was lied down and rolled multiple times at few hour intervals to facilitate mixing, and finally, it let sit for a few days for equilibration. No attempt was made to investigate the long-term drifts in this gas cylinder, e.g. due to wall adsorption and other surface effects, and therefore, we do not consider our approach as a reference method to be followed by others (and most notably, it does not fulfill SI-traceabliliy). We slightly modified our description of this procedure in Section 2.3 of the revised manuscript to address the Referee's concerns.

*"For this purpose, a regular steel cylinder (previously containing synthetic air) was evacuated, filled with synthetic air up to 1 bar and then a syringe with distilled water was used to introduce a given amount of $H_2O$, followed by pressurizing the gas cylinder with synthetic air (with $<5$ ppm $H_2O$ content, Messer, Switzerland) up to 100 bar, resulting in a humidity content of about 180 ppm $H_2O$. After a few days of equilibration, the gas was expanded into a 34 L SilcoNert®2000-coated stainless-steel cylinder (Essex Industries Inc, USA) to further minimize any potential surface effects during the measurements. It should be noted that this custom-made secondary reference gas does not fulfill SI-traceability, and it is subject to well-known long-term stability issues. Its sole purpose is to assess whether ALBATROSS is capable to measure significantly higher water vapor amounts than can be generated by the permeation method."*

P4, l28; it would also be good to specify the type of the SilcoNert®2000-coated stainless-steel cylinder into which the reference gas mixture was expanded. Despite the coating, surface effects might be different for different bottle sizes.

We used one 34 L Air Sampling Cylinder produced by Essex Industries Inc (St. Louis, USA). This information is now included in the text.

3.2 Pre-processing, P9, l2; I imagine that the spectral range was about 0.845 cm-1, so you obtain a spectral-point resolution of 1.69 10^(-4) for your stated 5000 datapoints? Please mention the spectral range to make clear how you got to the spectral-point resolution.

The details of the spectral range are already indicated in Figure S1. We have now specified the values in Section 3.2 of the revised manuscript as well.

*"The spectral range covered by the QCL is 0.88 $cm^{-1}$. The number of data points per spectrum is reduced by a factor of 4, i.e. from $2.1\times10^4$ to $5\times10^3$, by using the moving average approach, resulting in a uniform spectral-point resolution of $1.67\times10^{-4}$ $cm^{-1}$."*

3.4. Quadratic speed-dependent Voigt profile (qSDVP), p13, l14; I do not agree with the interpretation of the residuals obtained with the qSDVP as being free of any structure exceeding the normal noise level, although admittedly the features visible in Figure 6c and d are very small. Have you tried to see if an additional fit parameter (D2 != 0) or a higher order line shape model would be able to suppress this feature? If so, it would be nice to mention this. In any case, in my opinion it would be preferable to state that there is still some small structure observable, but that it is largely reduced compared to the VP. Stressing the argument that the QF obtained with the improved fit equals the SNR, one could nonetheless justify the choice of the qSDVP as line profile?

We appreciate this remark and realize that there is some contextual issue with this paragraph. The following text was added at the end of Section 3.4 in the manuscript to better describe the arguments for our decision regarding the selection criteria for the line profile parameters :

*"It can be tempting to include additional line profile parameter, e.g. $\Delta_2$ to further reduce the remaining structures. However, our primary aim is to find an optimum compromise between establishing a reliable and accurate spectral retrieval, while maintaining the high temporal (spatial) resolution of the spectrometer during balloon flights. The latter requires that we evaluate spectra from the flights at 1 s acquisition time rather than averaging them over, e.g., 50 s. However, as the noise scales by $\sqrt{t}$ (assuming random fluctuations), its amplitude is about 7 times larger in the 1 s data compared to the situation shown in the Figure 6, and thus, random noise-induced statistical effects dominate the spectra. In our opinion, including another degree of freedom for the spectral fit under such circumstances is largely questionable. Another aspect is the consideration of various artifacts and their impact on the measured line pro-files. As indicated by the Figure S1, the FSR determination uncertainty in our case is about $1.3\times10^{-5}$ $cm^{-1}$. Furthermore, the frequency stability of our free running QCL at longer time scales was found to be between 1.2 and $5.5\times10^{-5}$ $cm^{-1}$ (mainly determined by the laser heat-sink temperature stability). While the former term affects the frequency scale accuracy, the latter has a random bias on the line profile when averaging over multiple acquisitions. These influences can easily induce slight asymmetries or subtle line-shape distortions that can be than erroneously assigned to the $\Delta_2$ parameter. Moreover, our trial of using $\Delta_2$ as a free fitting parameter in the MSF routine resulted in a weakly constrained value with large uncertainties, indicating difficulties of a proper assignment. Similarly, we also found that considering other parameters, e.g. collisional narrowing, does not improve the QF index. Therefore, we conclude, in full agreement with previous works (e.g. Lisak et al., 2015), that the observed line*

*shapes can be well reproduced by assigning the non-Voigt effects to speed-dependent effects rather than collisional narrowing. These facts justify the choice of a reduced model against a generalized higher-order (HTP) parameterization."*

4.3. Linearity (extended-range validation), p18, l14; What about the repeatability of this measurement? Have you determined the ~180ppm concentration several times?

Due to the limited amount of the secondary reference gas (~1000 L), the measurement at ~180 ppm $H_2O$ (i.e., the undiluted secondary reference cylinder) could only be performed once. In particular, this measurement was performed at the end of the extended-range validation routine (i.e. after 64 hours of measurements). Therefore, it cannot be excluded that a slow decrease with time of the $H_2O$ amount fraction in the secondary reference gas (due to loss of $H_2O$ molecules to the walls of the cylinder) would indeed result in an underestimation of the initial $H_2O$ amount fraction. However, the magnitude of this drift can be estimated from the repeated (3×) measurements performed with diluted samples, e.g. at 140 ppm. To do this, we adopt a conservative approach and consider the statistical error on the undiluted reference gas measurement to be the sum of the standard deviation of the three individual measurements (repeatability) performed at 140 ppm (310 ppb $H_2O$), and the precision of the single measurement at 180 ppm (60 ppb $H_2O$). This results in a total statistical uncertainty of ± 370 ppb $H_2O$ (i.e., roughly ± 0.2 %) on the $H_2O$ content of the undiluted cylinder. This is now shown by the error bars on the ~180 ppm data point in Figure 9b. Furthermore, the following text was added to Section 4.3 of the revised manuscript to discuss this aspect

*"As the $H_2O$ content of the undiluted cylinder was only determined once, we estimate its uncertainty based on the repeatability of the measurement performed at 140 ppm (i.e., 310 ppb $H_2O$) and the precision of the single measurement at 180 ppm (60 ppb $H_2O$). This results in a total uncertainty of ± 370 ppb $H_2O$, i.e., roughly ± 0.2 % (as shown by the error bars in Figure 9b)."*

From Figure 9b it looks like all measurements (except for the 180ppm one) are slightly too high, as if the undiluted gas actually had a slightly higher water concentration…

Following this comment, we re-evaluated the dataset used for the linearity assessment and tried to find the reason for this deviation. It turned out that the expected $H_2O$ amount fractions were calculated using a wrong estimate of the $H_2O$ amount fraction in the undiluted cylinder (181.47 ppm), i.e., a preliminary result of a spectroscopic retrieval performed with exploratory conditions (namely, FSR = 0.024 cm$^{-1}$, $\Gamma_0$ = 0.0985 cm$^{-1}$ atm$^{-1}$, $\Gamma_2$ = 0.0155 cm$^{-1}$ atm$^{-1}$). The actual $H_2O$ amount fraction retrieved with the appropriate settings as given in the manuscript (FSR = 0.02429 cm$^{-1}$, $\Gamma_0$ = 0.0992 cm$^{-1}$ atm$^{-1}$, $\Gamma_2$ = 0.0135 cm$^{-1}$ atm$^{-1}$) equals 183.54 ppm. After recalculating the expected $H_2O$ amount fractions based on this value, we found that the systematic overestimation is actually removed, and all measurements lie now within ±0.6 % of their expected values. Figure 9, Table 4 and the text of Section 4.3 were updated according to the revised values and results. We thank the reviewer for this comment and apologize for the mistake.

Conclusion, p21, l10; "without systematic residuals", I do not agree, as discussed above.

Rephrased ("residuals" replaced with "bias").

**Technical corrections**

P9, l15, replace "while secondary reference gas mixtures (panel d) at 3 pressure levels (60−200 mbar)" by "while secondary reference gas mixtures (panel d) were measured at 3 pressure levels (60−200 mbar)"

Done.

P 14, l4, delete double "the" in "while the the qSDVP fitting uses"

Done.

The color-code used in the figures for different pressures (Figure 3, 4, 6, 7) should be unified. A different color scheme should be used, since the different blue lines are hardly distinguishable.

Done: the 250 mbar lines were changed from blue to black, and the same color-coding was applied consistently to Figures 3, 4, 6 and 7.

P 25, l15-26; references are not in correct order (K after L)

Order of references corrected.

---

## Author Comment (AC3)

We would like to thank the Referee for the thoughtful and constructive comments that helped us to improve our manuscript. We addressed each comment individually and made revisions in response to their suggestions, as detailed below. Our replies to the Referee comments are highlighted in blue. Modifications to the manuscript text are indicated in *italic*.

**Referee #3 (Daniele Romanini)**

Very nice work and paper, clearly exposed and overall rigorous in contents and discussions.

Your kind words are very much appreciated. Thank you.

First of all, I fully agree with all comments, questions and suggestions of the other two referees.

In particular, as already underlined, this works illustrates and confirms the interest of using laser absorption spectroscopy in particular in the mid infrared, to obtain high accuracy and selective measurements of atmospheric species. The presented instrument additionally possesses the uncommon feature of an open absorption cell which allows fast renewal of the atmospheric sample with minimal surface effects, especially valuable with sticky molecules such as water. This advantage is clear when looking at the long transient times (hours) observed when closing the cavity as is done for this study. About this point, it would be nice that the authors add a short discussion about the fact that the performance of the instrument can be considered the same when the cavity is open to ambient air. In particular, are there any turbulence effects which add noise to the measurements, or thermo-mechanical deformation of the optical setup which may increase measurement drift relative to what obtained here?

This is a key remark and a decisive aspect, and we are thankful to the Reviewer for raising this point. At the beginning, we asked ourselves exactly the same question and we spent quite some time in trying to establish a fact based verification of it. A detailed description of our finding was summarized in a recent publication (Tuzson *et al*., AMT, 13, 2020) demonstrating the performance of the QCLAS aboard drones. The environmental conditions (turbulence, temperature and pressure fluctuations, and mechanical vibrations) in such configuration are much more severe than flying aboard a weather balloon. We found that the highly reduced optical complexity of the QCLAS demonstrates an excellent robustness against environmental impacts and its performance was not deteriorated. The engineering solutions that allow to achieve such characteristics are detailed in the PhD thesis of Manuel Graf (see https://doi.org/10.3929/ethz-b-000429788). We added a short paragraph to Section 4.3 of the manuscript to summarize this aspect:

*"Although, these assessments were performed under well controlled laboratory environment, we expect that the performance of the instrument will remain the same also during flight conditions. The demonstration of this behavior is beyond the scope of this study, however, some related aspects (the effects of turbulence, thermo-mechanical deformation, etc.) were described in our earlier studies, e.g., Tuzson et al. (2020), Graf (2020), Graf et al. (2021). We found that the highly reduced optical complexity of ALBATROSS exhibits an excellent robustness against environmental impacts and its performance was not deteriorated."*

This work also illustrates that the Voigt line profile is largely inadequate for an accurate description of collisionally broadened molecular absorption lines. This is actually a well-known fact since the time that high-resolution spectra are being obtained by using narrow laser spectral sources - compared to spectra obtained at lower resolution with traditional Fourier or grating spectrometers. Nonetheless, it is very instructive to see the impact of the choice of spectral line shape on the retrieval of molecular mixing ratios, and to see that linear and accurate results may be obtained by using an advanced line shape model - with parameters determined by a multiline fit at several pressures, as shown in this work.

I have only one criticism concerning the long-term stability. The Allan-Werle stability analysis was performed for timescales only up to 500s (8 minutes). However, the importance of AW stability plots goes beyond the determination of the time for which a minimum AW deviation is attained, which defines the time for optimal averaging used in this work. The behavior of AW deviation at longer times provides essential information on instrumental drift. Do the measurements keep falling close to the optimal value or else do they drift away, and by how much and over how long? The AW deviation at long time scales provides this information on long-term stability of measurements and allows to assess the need for a re-calibration in case measurements are required to a level below the long-term drift. As it may be complicate to run an AW plot over more than one day, one can replace that by taking individual measurements averaged over the optimal time periodically and during a period which may be representative of the duration of a measurement campaign (one or 2 weeks).

We agree that longer measurement time would allow a more detailed insight about drift mechanisms acting at various time scales. However, in case of water vapor there is a particular technical difficulty related to the reference source. This has to be dynamically generated and the system involves multiple mass-flow-controllers, valves, and tubing. All these elements can, in principle, have long-term artifacts and thus affect the spectrometer performance. Nevertheless, we investigated a situation where we measured the dry synthetic air for up to ~1.5 hours. The results of the Allan-Werle variance analysis of these data is included now in the Supplementary Material as Figure S3. In this case, the spectrometer shows a precision of 0.5 ppb $H_2O$ reached after about 2 min averaging and then it remains constant over at least on a one-hour time-scale. We added the following text to Section 4.1 of the revised manuscript to highlight this aspect:

"*Investigating the time-series of the zero air measurements over longer time scales indicated that the spectrometer maintains a stable operation over a few hours, at least. The Allan-Werle deviation minimum in this case stayed at a constant value of 0.5 ppb $H_2O$ after about 2 min averaging (see Figure S3 in Supplementary material).*"

Drifts appearing at even longer time-scales, due memory effects in our sampling line and multipass cell (see discussion in the replies to Referee #1), were accounted for in the laboratory by re-measuring the empty-cell spectrum (i.e., dry synthetic air) every time before and after each experiment, which is then used to normalize the respective raw spectra. The stability is well demonstrated by the results of the accuracy validation (see Figure 8), which are based on repeated measurements of the same $H_2O$ amount fractions (at different pressures) over the course of ~5 days (8× experiments of 15 hours each).

Besides, in the present case, long term drift may explain some observed effects. For instance, if the measurement at 180 ppm used for determination of the amount fraction of the secondary reference

was performed a sufficiently long time before the measurements for the other amount fractions shown in figure 9, this might explain their systematic positive offset from the expected values.

The systematic overestimation of the measured $H_2O$ amount fractions in the linearity assessment was discussed in the context of the replies to Referee #2. This artifact was unfortunately due to a mistake in calculating the expected $H_2O$ amount fraction levels. Applying the right value, the systematic overestimation is removed, and all measurements lie within ±0.6 % of their expected values. Figure 9, Table 4 and the text of Section 4.3 were updated according to the revised values and results. We thank the reviewer for this comment and apologize for the mistake.

**Specific comments/questions:**

What is the finesse of the Ge etalon used for spectral calibration of the laser scans? Is it just an un-coated flat Ge slab, or does it have reflective coatings on its faces?

The finesse of the Ge etalon is very low, i.e. about 2.44. It is just an un-coated flat Ge slab. This limitation and its impact on the line profile is now discussed in the context of our replies to Referee #2 and in Section 3.4 of the revised manuscript accordingly.

Concerning the spectral resolution, the authors discuss briefly the "point resolution" and how that is fixed by the number of data points over a laser current scan, and their group averaging, however they should address the more fundamental question of the spectral resolution afforded by the line-width of the laser. Is it narrower than the point resolution?

For the icw-driver, we estimate a current noise density of 0.74 $nA/Hz^{-1/2}$. The typically observed 1/f laser noise PSD is ~$10^7$ $Hz^2/Hz$ at 1 kHz (see Liu et al., 2018). Hence, an approximate laser linewidth of about 650 kHz can be deduced. This about 8 times narrower than the point resolution that we used in our frequency scale. Considering its rather technical aspect, we decided to not include these details into this manuscript and keep the instrumental description compact.

Do they observe any excess noise in correspondence with the sides of an absorption line, before averaging several scans? At low pressure where the line sides are steepest, and at high H2O content for better S/N, excess noise relative to the spectrum baseline could reveal the effect and magnitude of the laser line-width.

We looked into the spectra recorded at 1 Hz using elevated $H_2O$ content (180 ppm) at the lowest pressure (60 mbar) that we used in the experiments, but we could not find any excess noise in the indicated range.

Any physical or practical reason for fixing Delta₂=0 in the fit?

Thank you for pointing out this weakness in our description regarding this aspect. We discussed this issue in detail in our replies to Referee #2 and added a more comprehensive description of the arguments for our decision regarding the selection criteria for the line profile parameters in the manuscript.

In table 1, the caption states that the relative uncertainty levels vary between 1.4-1.47% for all conditions but that does not seem to hold for the first point in the table (0.04/2.51=1.59%)

This is due to rounding the reference values given in Table 1 to the second digit after comma. The actual $H_2O$ amount fraction generated at this setpoint is 2.514 ppm with an expanded uncertainty of 0.037 ppm (i.e., 1.47 %). The full value with three significant figures is now given in Table 1. Thanks for noticing this issue.

---

## Author Comment (AC4)

We would like to thank the Referee for the thoughtful and critical comments that helped us to enhance the clarity of our manuscript. We addressed each comment individually and made revisions in response to their suggestions, as detailed below. Our replies to the Referee comments are highlighted in blue. Modifications to the manuscript text are indicated in *italic*.

**Referee #4 (anonymous)**

The authors present a strong, detailed, extensive study on a performance evaluation and optimization of their balloon-borne MIR-TDLAS-hygrometer ALBATROSS, which is definitely worth publishing.

Thank you for your positive feedback

**Introduction**: Due to the significant importance of the topic and the long lasting efforts of the airborne hygrometer community dating back way into the 1980s and 1990s I think the introduction should be revised to include essential representative and important work in airborne hygrometers, e.g., as FPH are mentioned as golden standard the review paper by Hall https://doi.org/10.5194/amt-9-4295-2016 should be relevant and mentioned. Further, the FISH hygrometer by C Schiller, reviewed by M Krämer in https://doi.org/10.5194/acp-15-8521-2015, is one of the most extensively used airborne hygrometers and one of the key reference instruments in AQUAVIT, and should be taken into account and mentioned.

Also there are plenty of airborne TDLAS hygrometers instruments e.g. by Sargent https://doi.org/10.1063/1.4815828, dating back in the 90s by Durry https://opg.optica.org/ao/abstract.cfm?uri=ao-38-36-7342 or Scott and Herman https://opg.optica.org/ao/abstract.cfm?uri=ao-38-21-4609. Particular relevant should be high flying ballon-borne open-path direct TDLAS hygrometers previously used for UT/LS sounding: *CHILD* by Gurlit et al https://opg.optica.org/ao/abstract.cfm?uri=ao-44-1-91 should be cited here. There is also work on new open-path hygrometers based on cMPC e.g. by Witt et al https://www.mdpi.com/2076-3417/11/11/5189 which I think should also be mentioned.

Particular stratospheric H2O vapor accessed via balloon platforms is strongly influenced by photochemical conversion of CH4 to H2O, asking for the need to simultaneously monitor traces of H2O and CH4, which is covered by some balloon sensors e.g. https://opg.optica.org/ao/abstract.cfm?uri=ao-44-1-91. A fact which should be considered and mentioned too.

We are well aware that there is a large amount of published work available about water vapor measurements in the upper atmosphere. In fact, the amount of research is so large that it would be valuable to write an up-to-date full review paper. However, this was not our aim, and therefore, we consciously limited the focus of this paper to "techniques demonstrated for lightweight balloon platforms" (page 2, line 15). Except for FPH by Hall et al. (2016), which is cited in the paper (page 2, line 12), none of the instruments mentioned above falls in this category. Although some of them were deployed on large atmospheric research balloons, their weight category is very different from our instrument, e.g: SDLA by Durry and Megie (1999), 20 kg; HWV by Sargent et al. (2013), 65 kg; ALIAS-II by Scott et al. (1999), 36 kg; CHILD by Gurlit et al. (2005), 20 kg; FISH by Schiller et al. (2015), 30 kg, while ALBATROSS (this work) is merely 3.5 kg.

Furthermore, we note that some of the suggested literature is already cited in our paper: besides Hall et al. (2016) (page 2, line 12), the FISH hygrometer is acknowledged through the review paper by Krämer et al. (2009) (page 2, line 5), and the most recent work by Durry and colleagues (Durry et al., 2008) about picoSDLA is also cited (page 2, line 17). Nevertheless, we underlined the large amount of available research in the field by adding the following paragraph to the introduction:

*"Since the pioneering work by Brewer and Dobson (1951), a large amount of scientific research has been done on the water vapor distribution and variability in the upper atmosphere, based on a wide range of platforms and analytical techniques (e.g., Scott et al., 1999; Rosenlof et al., 2001; Gurlit et al., 2005; Sargent et al., 2013; Buchholz et al., 2014; Meyer et al., 2015)."*

The dominant topic of the paper is hygrometer validation. The community has realized in the past a few fundamentally different type of validations: A) Field comparisons (Problem: lack of repeatability and lack of boundary parameter control) ; B) "Lab-like" parallel comparisons, e.g. AQUAVIT, (Problems: maintaining identical or at least comparable measurement and sampling conditions for all instruments and implementation of metrological references), and  C) Rigid single instrument validations, preferentially to a SI-traceable water vapor source (Problem: large total effort, lack of H2O sources suitable for atmospherically  relevant conditions, i.e. accurate definition of trace H2O levels - variable low gas pressures - low air temperatures). These differences should also be part of the introduction in order to avoid comparing "apples with oranges". In addition to AQUAVIT other validations of the above categories should cited / taken into account/ analyzed, e.g:

Buchholz + Smit > field comparison https://doi.org/10.1007/s00340-012-5143-1;
Filges + Gerbig > field comparison  https://doi.org/10.5194/amt-11-5279-2018;
Buchholz + Ebert > metrological standard https://doi.org/10.5194/amt-11-459-2018  ;
Buchholz > metrological primary standard https://doi.org/10.1007/s00340-014-5775-4

For AQUAVIT (which was the largest parallel hygrometer comparison under variable p-T-H2O conditions), the authors should not highlight the (insufficient) performance of very young - not matured – instruments ( "exceeding 100%") and give their underperformance the same weight like the very mature CORE hygrometers, which have been used and improved over decades. The performance of the non-calibrated, absolute, open-path TDLAS "APicT" in AQUAVIT certainly also relates well to the paper here, and could be mentioned in the paper. Some of the main findings of AQUAVIT were indeed the still quite large total discrepancies (-+ 10% relative) between the very mature "core" hygrometers. Also it took a complicated decision making process to define a "comparison reference" i.e. a suitable metrological H2O source or metrologically validated reference instrument which is compatible with the special (low temperature) boundary conditions of the AIDA chamber and their huge size.

We fully agree that the intercomparison of the measurement techniques is a very challenging issue. Even more, we have gained extensive knowledge and experience in this topic as we recently participated in the last AquaVIT-4 intercomparison campaign that took place at the AIDA chamber in April 2022. We are preparing a separate manuscript dealing with all the aspects of such intercomparisons, and thus we focus our current paper on the assessment of our spectrometer.

We believe that a sufficient level of detail is given to the reader regarding the different methodologies employed by the intercomparison papers, which are cited. In particular, concerning the discussion of the AquaVIT-1 intercomparison results (Fahey et al., 2014), we note that both aspects underlined by the referee ("core" instruments within ±10 %, other instruments exceeding ±100 %) are in fact already mentioned in the manuscript (see page 2, lines 9-11).

**Experimental**:

The authors target a SI-tracebale validation of ALBATROSS, where ALBATROSS is claimed to be a gas sampling-free, and calibration-free open-path Mid IR spectrometer.

The open-path approach promises to avoid H2O adsorption problems. However, open path also causes a very complicated tradeoff in system design, due to the complete lack of "sample

control" during field-use, so that gas pressure, gas temperature, residence time, sample homogeneity must be measured, evaluated or assumed. Additionally p and T are often not measured within the optical sample volume but outside of it, leading to further heterogeneity errors in measured p and T wrt to the gas sample.

In my understanding a validation of the open-path ALBATROSS was not tackled or described in the paper. Instead a closed-path version of ALBATROSS was used, which I think, is a big change with respect to the initial claim. Of course, the validation of the closed-path version is highly important and demanding, but closed-path studies are certainly not fully sufficient to validate the open-path version and certainly not under UT/LS field conditions. The title of the paper is therefore misleading and should be considered to be changed ( e.g. > *"validation of a closed-path Albatross"*).

We share many thoughts of the referee, and we are well aware of the challenge given by the fact that the "perfect" validation does not exist (see options A – C above, described by the referee). Therefore, we have adapted the title to clearly state that the validation was not done ON a balloon-borne spectrometer but FOR a balloon-borne spectrometer, i.e.:

*"SI-traceable validation of a laser spectrometer for balloon-borne measurements of water vapor in the upper atmosphere"*

Nevertheless, we would like to note that it is common practice to determine spectroscopic parameters in a close-path system and apply them to open-path measurements. In fact, this is the scientific ground for all remote (including satellite) measurements, and there is no need to fundamentally question this concept.

Our laboratory experiments in a closed-path, but otherwise identical, configuration are likely the best, or even only, approach to characterize the spectroscopic performance. We see no reason to doubt its validity for the open-path configuration, and the paper makes it clear that the experiments are done in closed-path configuration (proof given by the comments of the referee).

The other technical aspects mentioned by the Referee, i.e. gas pressure, gas temperature, residence time, and sample homogeneity apply to all field instruments, regardless of the measurement technique. In fact, it is hard to think of a setup that is more representative and resilient than ALBATROSS under slow flight conditions and in the UTLS. Nevertheless, this has been (Graf et al. 2021) and will be (AquaVIT-4, manuscript in preparation; further in-flight comparisons) discussed in more detail. This is discussed in the conclusions, making it clear that this is not the last, and likely not the final assessment. Adding a very extensive discussion here would add more noise than weight to this paper, which we consciously kept in a focused manner.

In order to deduce the performance of the open-path version careful consideration and evaluation of p, T sensor location and calibration is needed, which is however not given in the paper. Witt et al in https://www.mdpi.com/1424-8220/23/9/4345 recently evaluated a comparable open-path C-MPC under dynamic situations and found considerable systematic deviations caused by spatial gas temperature inhomogeneity and by the un-even statistical spatial weighting caused by the special C-MPC beam pattern. This findings are probably of high relevance for open-path in-field-use as well as for high-accuracy validations in closed-path cMPCs as presented in this manuscript by the authors.

We agree that p and T measurement in an open path configuration during flight is a topic under debate and up to know there is no 'golden'-solution for solving this issue. This aspect is an inherent problem to all measurement techniques, independently of the platform used. Thus, efforts as described by Witt et al. are very useful studies and deliver important insights in

optimizing p and T measurements (especially for sampling aboard airplanes), applying the necessary corrections or estimating the uncertainties.

In our paper, we explicitly communicate that the assessments are done under well-controlled laboratory conditions where these parameters are kept constant and homogeneous.

For the sake of completeness, we note that the paper by Witt et al. (2021) is dealing with the evaluation of gas temperature and concentration inhomogeneities in dynamic tube flows. While doubtlessly an important investigation, the targeted application there is rather different from the conditions of a balloon flight, not to mention the low-flow and constant temperature conditions used in our laboratory assessment. Even in the atmosphere during flight, the temperature and the water vapor concentration are locally much more homogeneous than the conditions investigated in the Witt *et al* paper, i.e. "strongly heterogeneous T fields generic for industrial process application, e.g., in pipe flows" and "temperature range from 293 to 473 K at 1 atm of pressure."

Interestingly, Witt *et al*. found, despite the harsh conditions, that "for the case of a strong thermal boundary layer with a delta-T of 180 K (…) would lead (…) to a relative deviation of −5.3 % between the "true" and the calculated concentration." Considering our well-stabilized setup, with $\Delta T < 0.1$ K, the impact of such effects on the accuracy is negligible.

The authors aim on calibration-free first-principles evaluation of the hygrometer signals, which is indeed a very powerful capability for field use (see e.g the airborne HAI Hygrometer). In a cal-free mode, however, the TDLAS-instrument integrates any $H_2O$ spectral absorption over the full light path i.e. anywhere between the laser chip and the detector chip. Any "parasitic" = unwanted water along the absorption path "outside of the absorption cell" will lead to systematic, potentially drifting offsets and needs to be carefully evaluated and removed. Particularly complicated are situations where the gas pressure also is heterogenous along the path (e.g in sealed laser or detector housing). This problem is carefully described in Buchholz 2014 https://iopscience.iop.org/article/ 10.1088/0957-0233/25/7/075501. How this is solved / or avoided in the present study must also be described, in particular if ALBATROSS is claimed to be cal-free. It is unlikely that this problem is completely absent in the ALBATROSS design. Parasitic water vapour offsets can of course be removed to first order by calibration, but not in a cal-free TDLAS hygrometer.

This is a very valuable and important remark. The "parasitic" absorptions along the free space path are indeed challenging. Our solution is to keep the distance between laser-MPC and MPC-detector as short as physically possible. In our case, this amounts to 2.7 cm. This short path is then enclosed by a flexible tubing that is purged with dry $N_2$ such that the absorption contribution from the residual water drops below the detection limit of the instrument. The technical details of this solution will be discussed in our next paper, describing the flight measurements, where the influence of this artifact can have a substantial impact, mainly due to the similar pressure conditions within and outside the SC-MPC.

We added the following clarifying text to Section 2.1 of the revised manuscript:

"*Furthermore, the free-space path between the key optical elements, i.e. laser-MPC-detector (kept by design as short as physically possible, in our case, 2.7 cm) was enclosed by a flexible PEEK-tubing that is purged with dry $N_2$ and maintained slightly above atmospheric pressure, to avoid any "parasitic" $H_2O$ absorption from these external path sections.*"

With respect to this topic it should also be analyzed where the zero air blank values (e.g. 1.46 ppm in Fig 2 compared to 0.59 ppm in fig 9 ) comes from, how stable they are and e.g how much of this is caused by parasitic water in the spectrometer itself.

The non-zero H$_2$O content (~1.5 ppm) observed in the laboratory while measuring dry synthetic air originates exclusively from the memory effects of the setup, mainly tubing surfaces and the large surface-to-volume ratio of the closed SC-MPC (see also the replies to Referee #1).

As mentioned above, the free-space OPL of 2.7 cm was efficiently purged and maintained slightly above atmospheric pressure to avoid "parasitic" H$_2$O absorption from these external path sections. The large pressure difference between inside and outside SC-MPC would also allow for an easy separation of the two contributions during the spectral fitting. Because of this, "parasitic" water in the spectrometer itself can be definitely excluded. Nevertheless, we acknowledge that the above description of the purged design was missing, and the Referee's critique was therefore well motivated.

In the spectral evaluation a "spectrum normalization" via a division through an "empty cell spectrum" is used. As the "empty cell" still had non-stable "zero" water levels of 1,5 ppm, the spectrum normalization actually also introduces an effective offset (and to a certain extend a parasitic water vapor ) correction. This approach and the offset correction cannot be used in the open-path configuration. The alternative approach "polynomial baseline reconstruction" does not provide offset correction so that the parasitic contributions  should be effective. The authors should add data on this if possible or discuss this  effects and their quantitative influence on the absolute accuracy of  both ALBATROSS  versions.

The empty-cell spectrum normalization procedure applied to the laboratory validation data is correctly described by the referee. However, this offset correction is not required in the open-path configuration (i.e., under flight conditions), where there is no sampling line, the gas flow is much larger, and the surfaces of the SC-MPC are drastically reduced, as the lids are removed and the gas-surface interaction is limited to the narrow inner circumference of the cell. This aspect was already discussed in a reply to Referee #1, and a clarifying sentence was added to Section 2.4 of the revised manuscript.

**Used preparative water vapor references:**

**Primary Permeation source:** It should be better clarified which components in the entire setup (fig 1) define the "SI-traceable permeation source". Is this the permeator only, or the permeator and  MFC1 and 2, or even more components? This needs to be clarified as only this subsystem provides the property of being SI-traceable. As I see it now, the "permeation source subsystem" is embedded into a larger gas mixing system containing further MFCs plus an additional pressure controller(s?), pressure sensors, gas and cell body temperature measurement. These all should be shown on fig 1. and better explained in the text. For the entire validation to be "SI traceable" all relevant measurement data need to be SI traceable. Traceable calibration data and accuracy and expanded uncertainties should be provided for all measurement parameters (p, T, flow etc) required for the TDLAS evaluation procedure, which is not the case. Figure 1 lacks also an excess flow outlet before MFC3.

In Figure 1, the SI-traceable part of the magnetic suspension balance (MSB) is indicated by the grey shaded area. This entire unit represents the core of the metrological-grade solution used at METAS and it was validated in several intercomparison studies.

The overall uncertainty of the spectroscopic retrieval was already given in Section 3.1 of our manuscript.

The absolute accuracy and stability of the reference H2O concentration and the gas handling system will influence the TDLAS validation and e.g. depends on accuracy and stability of the H2O blind value, which needs to be determined and should be given in the text.

As we used a solely spectral retrieval (i.e. purely deduced from molecular and environmental parameters), none of the spectroscopically determined values were calibrated or linked to the SI-traceable values. These latter data were used for comparison purpose only. Therefore, each value represents basically a blind value. The absolute accuracy and stability of the generated reference $H_2O$ concentration is already discussed in Sect. 2.2 in our manuscript.

Due to the lack of traceability information for the used validation setup I can't see that the "entire validation setup" is SI traceable. Due to this deficit, the paper claims and the title should better changed to, e.g *"Validation of a closed-path balloon-borne spectrometer with a permeation-based SI traceable H2O-source"*.

As explained above, the source is traceable, and so are all other elements upstream of the spectrometer. The corresponding uncertainties are also given in the text. The referee is well aware (see discussion of types of validation) that there is no such thing as an artificial, traceable stratospheric chamber in which an open-path instrument can be flown on a balloon. Within this fundamental constraint, the high-level and traceable experiments described in the manuscript are adequate. The main limitations are discussed in the text (with additional notes following the valuable comments of the referee), and further validation steps (e.g. flights and chamber measurements) are mentioned in the conclusions. Overall, we believe, that this gives sufficient and balanced information to the reader.

**Secondary water standard:** The bottled H2O mixture generated is analyzed (if I got it right) only by the closed-path ALBATROSS. Hence the assigned bottle value of 181 +- 0.06 ppm "collects" all uncertainties (and all systematic errors) from the closed-path Albatross validation using the primary permeation source. The +-0.06 ppm (=3,3 E-4 relative!) can thus only be "precision". Here the accuracy and the uncertainty of the bottle assignment should be added and discussed, which then needs to be taken into account for the "expected H2O amount fraction" in fig 9, and for the evaluation of the uncertainty of the linearity relation. Looking at the fitting function in fig 9 the differential linearity and the 1,008 slope seem certainly excellent. However, the large offset of 590 ppb (which is very close to 2% ! at 30 ppm and would extrapolate to 15% at tropopause concentrations of 4 ppm) definitively needs further explanations by the authors. For me this indicates an accuracy problem of closed-path ALBATROSS or/and this secondary standard setup. Also for both values (m and b) uncertainties should be provided.

Other points to be considered are the likely dependance of the H2O amount fraction form the bottle pressure, as well as sampling influences by the sampling line including the pressure reducer, more information on the sampling system and the adsorption minimization would be helpful.

The systematic overestimation of the measured $H_2O$ amount fractions in the linearity assessment was discussed in the context of the replies to Referee #2. This artifact was unfortunately due to a mistake in calculating the expected $H_2O$ amount fraction levels. Applying the correct value, the systematic overestimation is removed, and all measurements lie within ±0.6 % of their expected values. Accordingly, the linear fit results are also improved. Figure 9, Table 4 and the text of Section 4.3 were updated to the revised values and results. We thank the referee for this comment and apologize for the mistake.

Taking all this into account, bottle-based secondary standards might be useable as a high concentration H2O source, but to be useful to a broader community they certainly need more evaluation work.

The secondary water standard does not fulfill SI-traceability, and it is subject to well-known stability issues. Its sole purpose is to assess whether ALBATROSS is capable to measure significantly higher water vapor amounts than can be generated by the permeation method (see also the replies to Referee #2). We have added a sentence to Section 2.3 of the revised manuscript to further clarify this aspect:

*"It should be noted that this custom-made secondary reference gas does not fulfill SI-traceability, and it is subject to well-known long-term stability issues. Its sole purpose is to assess whether ALBATROSS is capable to measure significantly higher water vapor amounts than can be generated by the permeation method."*

**Spectroscopic retrieval:**

The spectroscopic retrieval section is quite extensive and specialized for publication in AMT.

In my view the full fitting model is not sufficiently described: It is not clear how many and which water lines (or other interfering species) are fitted, or e.g how large the pressure dependent influence from neighboring lines is and how and if it is compensated. Which H2O isotopic composition is assumed? An H2O stick spectrum showing the fitted as well as the ignored lines would be helpful here. Also the description of the physical model behind the spectral evaluation and in particular a complete set of input parameters and their total uncertainties is not given. A total uncertainty evaluation of a cal-free system seems therefore not possible.

The employed quadratic speed-dependent Voigt profile (qSDVP) is well established and recommended by IUPAC. The spectral line (1662.809 cm$^{-1}$) is given in the manuscript. At the relevant pressure (< 250 mbar) and concentration (<35 ppm) there is no spectral interference from neighboring water lines and, thus, the fit considers only the isolated line. The closest and most relevant line would be from H$_2^{18}$O (181) at 1662.3353 cm$^{-1}$ with an absorption amplitude of $3\times10^{-4}$ that has no impact on our retrieval.

The line profile parameters used in our study are listed in Table 2 and illustrated in Figure 5.

The selected absorption line belongs to the main water isotope (161). The isotopic composition of our water standard is not known, but it can be assumed that of typical tap water, i.e. ~-60 ‰ $\delta^2$H and -10 ‰ $\delta^{18}$O. Considering the low abundance of the heavy isotopologues (0.27 %), the total estimated uncertainty would be about ± 0.02 %. This additional uncertainty has been introduced in Section 2.2 of the revised manuscript, and the reference levels generated by the dynamic-gravimetric method were scaled by a factor 99.73 %.

*"As the isotopic composition of our water standard is not known, we estimate an additional uncertainty of about ± 0.02 % on the total H$_2$O amount fraction, by assuming that the liquid water standard has a signature of typical tap water, i.e., −60 ‰ $\delta^2$H and −10 ‰ $\delta^{18}$O. Considering the low abundance of the heavy isotopologues and a natural distribution, the total amount fraction contains about 99.73 % of the light water isotopologue (H$_2^{16}$O). Since ALBATROSS measures only this water isotopologue species and not the total H$_2$O amount fraction, the reference values generated by the dynamic-gravimetric method were scaled by this factor for the accuracy assessment."*

Consequently, the results of the accuracy assessment (i.e., Figure 8, Table 3 and the text of Section 4.2) were updated in the revised version manuscript, after rescaling the reference levels generated by the dynamic-gravimetric method by a factor 99.73 %. The main conclusions of the accuracy assessment are unaffected by this change. We thank the referee for this valuable remark.

If the cal-free evaluation is the goal, then all spectral parameters plus all auxiliary measurements needed (= p, T, L ….) must be stated with their (expanded) accuracies/uncertainties. Here I would expect an uncertainty table for all input parameters, as well as more information on p-, T-sensors their location and traceable calibration, which is not given.

See our reply above. The uncertainties of the *p* and T measurements are already given in the manuscript (see Section 2.3).

Also the uncertainty influence of the fitting process itself as well as e.g. the uncertainties of the linearization of the spectral axis/laser tuning should be discussed.

We agree. This aspect and especially the impact of the uncertainty of the laser tuning is now discussed in more detail in Section 3.4 of the revised manuscript (see also our replies to Referees #2).

Particularly in gas spectrometers the real gas temperature in a weakly thermally conducting low pressure gas can cause problems. Concerning the HMP110 used here: this T sensor is specified by the manufacturer with an accuracy of 0.4K (not 0,2K) for the extended T range (needed for UTLS use). Also comments by the authors are recommended if/how they deal with the systematic T-offsets /uncertainties (and the effect in the TDLAS evaluation) caused by invasive air temperature measurements, i.e. evaluation of PT100's self-heating and thermal gas to sensor transfer problems (particularly at low pressure). EURAMET project 1459 "Air Temperature Metrology – ATM" could be considered here.

Again, the overall uncertainty of the spectroscopic retrieval has been specified in our manuscript (see reply above). The largest source of error is the permeation source (1.5 %), while the absolute uncertainties on the measured p (0.12 %) and T (0.06 %) play a secondary role. Concerning the HMP110, we consider only the conditions that are representative for our assessments, i.e. room temperature, where the value of 0.2 K applies. For the flights, we use other calibrated temperature sensors, which are much smaller than the HMP110. For the sake of clarity: in-flight uncertainty may be different, e.g. because of such small adaptations and because of other effects, such as contamination by the balloon wake (Graf et al., 2021). This is quite fundamental to any validation approach as the reviewer rightly pointed out at the very beginning.

The magnitude of the "temperature problem" also strongly depends on the spectral line selection: H2O line identification and lower state energy of the fitted lines therefore should be given in the paper.

The selected line has a lower state energy of 79.5 cm$^{-1}$, which makes it largely insensitive to the "temperature problem". For example, a temperature change of 1 K would give < 0.13 % change in the observed absorption amplitude. Since we take this effect into account in our calculations, and because the ambient temperature is maintained constant, the impact of this term can safely be considered negligible. The following sentence was added to Section 2.1 of the revised manuscript:

*"This absorption line corresponds to the transition 221←212 with a lower state energy of 79.5 cm$^{-1}$, which makes it largely insensitive to temperature. For example, a temperature change of 1 K would give < 0.13 % change in the observed absorption amplitude. Since we take this effect into account in our calculations, and because the ambient temperature is maintained constant, the impact of this term can safely be considered negligible."*

The influence of individual spectral data uncertainties can be quite large and often strongly limits the achievable total uncertainty of cal-free spectrometer realization. As I understand the authors paper, they are taking fixed line strength S(296K) and broadening $G_0$ for the spectral evaluation from HITRAN, further they need T dependance of broadening and S (which also comes from HITRAN with their uncertainties) and the line pressure shift Do (again HITRAN + uncertainty) and then finally the "new" qSDVP braodening parameter $G_1$ (which also needs to get an uncertainty from the parametrization). With typical HITRAN uncertainties for S 1-10 % (depending on the line selection) Voigt broadening (another U= 2-5 % ) , broadening T coefficient (5 -20% and more ), plus p, L, T, fit process, tuning and spectral axis uncertainties it certainly takes further explanations how the closed-path ALBATROSS reaches 1.5% total uncertainty in cal-free mode. The best short-term accuracy can certainly be achieved by a hygrometer calibration to a very good reference and not via a spectroscopic cal-free approach, due to the large amount of spectral input parameters with fairly large uncertainties.

The estimated uncertainty of the line strength for the selected transition is specified in the HITRAN database with code 7, corresponding to a relative uncertainty $\geq 1$ % and $< 2$ %. This represents the largest uncertainty of the spectroscopically retrieved values. This information was now added to Section 2.1 of the revised manuscript.

We demonstrate an agreement between the SI-traceable sample and the spectroscopically retrieved (measured) values within the uncertainty (1.5 %) of the sample when using the qSDVP profile with the parameters given in Table 2. As discussed below, please note that the qSDVP broadening coefficients are determined by letting the MSF minimize the fit residuals for all the pressure levels at once, i.e. this is the only criteria to determine the line-profile parameter values which are thus independent of the absolute value of the water vapor source.

Nevertheless, for the sake of clarity, we decided to not use the term "calibration-free" in our manuscript, but simply state the fact that the measurements show that ALBATROSS achieves an accuracy better than $\pm 1.5$ % with respect to the SI-traceable reference at all investigated pressures and $H_2O$ amount fractions. In this context, it may be valuable to recall that accuracy is also understood as the closeness of agreement between measured quantity values that are being attributed to the measurand (see e.g. VIM 3).

The transfer of the closed-path validation presented in the paper to the open-path balloon-version, depends particularly strong on the accuracy of the spectral data i.e. H2O line selection or the temperature coefficients of the broadening. What measurements this requires and how this should be described is shown e.g in Pogany et al for traceable H2O strength https://doi.org/10.1016 /j.jqsrt.2015.06.023 and in Nwaboh https://doi.org/10.3390/app11125341 for traceable determination of H2O broadening incl T dependence for TDLAS.

The only parameters not covered by our study are the uncertainties of the temperature dependency of the line strength and the temperature coefficient of the broadening parameter. This limitation is clearly stated in our paper. As mentioned above, we have now added supporting information about this in Section 2.1 of the revised manuscript.

Although, this does not apply to our measurements performed at room temperature, temperature dependency of the line strength and the temperature coefficient of the broadening parameter might have an impact on the atmospheric water vapor measurements in the UTLS. However, their effect is still expected to be of minor compared to the uncertainty (~1 %). of the line strength.

**Line shape study:**

The authors compare the applicability of two line shape models: Voigt (VP) and quadratic speed dependent Voigt (qSDVP) and then optimize the qSDVP approach. Their VP evaluation is not very extensive and based on a single fixed set of parameters taken from HITRAN: The to be expected pressure dependent line shape deficits are not taken care of. It should be noted that Buchholz AMT 2018 had proposed to correct this parametrizable, perfectly long-term stable, systematic deviation caused by the Voigt profile deficits e.g via a look-up table approach. This correction approach allows faster fitting and avoids too many fit parameters, which have caused in his spectrometers noise-like fitting instabilities.

This paper is not primarily a study of line-shape models, and more extensive studies on VP have been published. Nevertheless, we give a very thorough insight that will be appreciated by the community (see e.g. the three previous reviewers), e.g. a) we clearly show the VPs difficulty to properly describe the pressure effects (see Fig. 8), b) we show that with "fine-tuning" of $G_0$ and lines strength an excellent agreement can be obtained, but the pressure-bias would still dominate the uncertainty (see Fig.4 and consider the case $G_0 = 0.0954$), and c) we indicate the fit residuals and their dependency on pressure and $H_2O$ concentration on Fig. 6a,b.

The look-up table approach, proposed by Buchholz et al., although computationally efficient, relies on normalization factors, which have no physical meaning and may strongly depend on instrumental parameters. We avoid such an approach and rely on physically motivated and well-studied quantities instead.

For the qSDVP evaluation the authors use a restricted qSDVP parameter set, allowing only one additional "broadening parameter", and hence should be better called "simplified qSDVP" to be precise. As I understand the paper, the authors use the Albatross-permeation standard-comparison via an iterative parametrization to "determine" the "optimal" qSDVP broadening parameters for their setup (while checking S). The parametrization of the width parameter of the simplified qSVDP follows two goals A) to match the spectrometer response function (Albatross H2O concentration) and the reference concentration (permeation source plus mixing system) and B) to minimize the fit residual i.e. to remove systematic deviations in the line shape fitting (optimization of QF). However, there is no uncertainty provided for the outcome of this process, leaving it open, how accurate this qSDVP parameter really is. Literature comparisons of this spectral parameter are also not given, making it somehow an instrumental parameter.

As the goal of the parametrization was to improve the "correlation" between the permeator and the TDLAS system, it is "no surprise" that the parametrized qSDVP evaluation yields pretty much the "input data set", while the QF optimization improves the apparent system precision by minimizing the fit residual. However, the problem I see is, that the reference permeator information was used twice: First for the determination of the spectral information and then the "trained" qSDVP-TDLAS was compared to its previous reference in the learning situation. And the result of the second step is not really surprising, it's a pretty good match.

Obviously, there is a fundamental misunderstanding of our approach. We do not use the permeator values in our spectral retrieval at any stage. The line profile parameters are optimized by the fitting routine using the Levenberg-Marquardt least squares algorithm. The only criteria for selection of the line profile parameter values is the fit residual from the MSF, i.e. when the highest QF is obtained. The "real" $H_2O$ concentration is not used anywhere in the fitting process as a constrain or "training" set. The model is based on first principles using exclusively molecular parameters and environmental/physical (p, T, OPL) values to describe the observed absorption signal.

In our opinion, it is staggering to find such an outstanding match between the purely spectroscopic values and the "real" $H_2O$ concentration values. This indicates the high quality

of the molecular parameters listed in the HITRAN database and their well confined uncertainty, at least for the H$_2$O transition selected for our study. Furthermore, it also demonstrates that beyond-Voigt line-profile models have the capability to accurately describe the observed shape of the absorption line under varying pressure conditions. We added the following paragraph to our manuscript to strengthen this point:

"*It is important to realize that the line profile parameters are solely determined by the QF. The MSF algorithm is not aware of the target (or "true") value of the H$_2$O concentration, it simply tries to minimize the sum of the squares of the residuals, i.e. the difference between the observed and the fitted value provided by the model. Here, the model is based on first principles using the molecular parameters and the physical quantities (p, T, OPL). The generated SI-traceable H$_2$O concentrations are used for comparison purposes only. There is no calibration involved.*"

Furthermore, we extended our discussion to include the important finding:

"*This excellent agreement reflects the high quality of the molecular parameters listed in the HITRAN database and their well confined uncertainties, at least for the H$_2$O transition selected for our study. Furthermore, it also demonstrates that beyond-Voigt line-profile models have the capability to accurately describe the observed shape of the absorption line under varying pressure conditions opening the path to a highly accurate quantification of the observed data.*"

For me this approach seems essentially like a more elegant way to calibrate the spectrometer response function by using the reference H2O concentration. The uncertainties of this process are not sufficiently discussed. Also the high correlation caused by that approach is not studied or taken care of. An elaboration of this problem would require a further comparison with a third independent preparative or analytical H2O system, which has not been shown in the paper.

Therefore I think that the authors cannot claim a demonstration of a calibration-free hygrometer. Not in closed-path configuration and even less so with an open-path cell.

See our reply above. Our spectral evaluation is not correctly understood and, therefore, the conclusion is not appropriate. We do not apply any calibration, but simply compare the spectroscopically derived values with the SI-traceable ones. The main uncertainty is given by the reference material. In principle, the spectrometer would be able to achieve accuracies that are of similar level than its precision.

As we explained above, the close cell configuration is actually more challenging as it involves surface and memory effects that are not present in the open cell case. The physics of light-matter interaction is independent of the MPC configuration. We are confident that our assessment is similarly valid for both cases, but we acknowledge that during flight, additional factors may have an impact on the accuracy. This is, however, related to the field conditions and not to the assessment and validation steps as presented in our study.

Essentially they have developed a novel (?) calibration procedure instead. In contrast to a classical calibration they are not aiming on a direct correction of the instrument function but realized a "physics-informed approach" to remove line shape deficits. This is also valuable(!) but it remains a calibration process. Also I think further work is needed to investigate the accuracy and (longer term) stability of this parametrization / parametric calibration, and how often it needs to be repeated. But I think that the claims derived from the data and the results should be carefully and conservatively revised. What I see is a "Use of a SI-traceable permeation source for the characterization/calibration of a closed-path Mid-IR QCL TDLAS hygrometer suitable for balloon-borne, extractive UTLS-hygrometry"

Once again, we use well established and recommended (see IUPAC) line shape models to best capture the observed line profile. The line parameters are obtained by applying standard

mathematical methods (least squares fitting) to optimize the residuals between the observed and calculated line profile. At this point, we would like to cite the remark of Referee #3: "This work also illustrates that the Voigt line profile is largely inadequate for an accurate description of collisionally broadened molecular absorption lines. This is actually a well-known fact since the time that high-resolution spectra are being obtained by using narrow laser spectral sources (…). Nonetheless, it is very instructive to see the impact of the choice of spectral line shape on the retrieval of molecular mixing ratios, and to see that linear and accurate results may be obtained by using an advanced line shape model - with parameters determined by a multiline fit at several pressures, as shown in this work."

It is unfortunate, that apparently we were not able to clearly communicate our spectral retrieval. We take the opportunity to improve this aspect in our revised manuscript correspondingly (see also our replies to the other referees). As the values retrieved by ALBATROSS are exclusively relying on first principles, molecular parameters and environmental/physical (p, T, OPL) values, their comparison to the SI-traceable values is the best available way to show their quality. Furthermore, we are preparing another manuscript where we show intercomparison results from balloon flights using the CFH as reference. This will demonstrate the capabilities of ALBATROSS also in the open-path configuration under flight conditions.

Nevertheless, we made a slight change to the title of our manuscript to accentuate even more that the assessment was not done ON a balloon-borne spectrometer but FOR a balloon-borne spectrometer:

*"SI-traceable validation of a laser spectrometer for balloon-borne water vapor measurements in the upper atmosphere"*